# Small coresets via negative dependence: DPPs, linear statistics, and concentration

**Rémi Bardenet** [*]
Univ. Lille, CNRS, Centrale Lille,
UMR 9189 – CRIStAL,
F-59000 Lille, France
`remi.bardenet@cnrs.fr`

**Subhroshekhar Ghosh** [*]
Department of Mathematics
National University of Singapore
10 Lower Kent Ridge Road, 119076, Singapore
`subhrowork@gmail.com`

**Hugo Simon-Onfroy** [*]
Université Paris-Saclay, CEA, Irfu
Département de Physique des Particules
91191, Gif-sur-Yvette, France
`hugo.simon@cea.fr`

**Hoang Son Tran** [*†]
Department of Mathematics
National University of Singapore
10 Lower Kent Ridge Road, 119076, Singapore
`hoangson.tran@u.nus.edu`

## Abstract

Determinantal point processes (DPPs) are random configurations of points with tunable negative dependence. Because sampling is tractable, DPPs are natural candidates for subsampling tasks, such as minibatch selection or coreset construction. A *coreset* is a subset of a (large) training set, such that minimizing an empirical loss averaged over the coreset is a controlled replacement for the intractable minimization of the original empirical loss. Typically, the control takes the form of a guarantee that the average loss over the coreset approximates the total loss uniformly across the parameter space. Recent work has provided significant empirical support in favor of using DPPs to build randomized coresets, coupled with interesting theoretical results that are suggestive but leave some key questions unanswered. In particular, the central question of whether the cardinality of a DPP-based coreset is fundamentally smaller than one based on independent sampling remained open. In this paper, we answer this question in the affirmative, demonstrating that *DPPs can provably outperform independently drawn coresets*. In this vein, we contribute a conceptual understanding of coreset loss as a *linear statistic* of the (random) coreset. We leverage this structural observation to connect the coresets problem to a more general problem of concentration phenomena for linear statistics of DPPs, wherein we obtain *effective concentration inequalities that extend well-beyond the state-of-the-art*, encompassing general non-projection, even non-symmetric kernels. The latter have been recently shown to be of interest in machine learning beyond coresets, but come with a limited theoretical toolbox, to the extension of which our result contributes. Finally, we are also able to address the coresets problem for vector-valued objective functions, a novelty in the coresets literature.

## 1 Introduction

Let $\mathcal{X} = \{x_i \mid i \in [\![1, n]\!]\}$ be a set of $n$ points in a Euclidean space, called the *data set*. Let $\mathcal{F}$ be a set of nonnegative functions on $\mathcal{X}$, called *queries*. Many classical learning problems, supervised or

---

[*] The authors are listed in alphabetical order by their surnames

[†] Corresponding author

38th Conference on Neural Information Processing Systems (NeurIPS 2024).

unsupervised, are formulated as finding a query $f^*$ in $\mathcal{F}$ that minimizes an additive loss function of the form

$$L(f) := \sum_{x \in \mathcal{X}} \mu(x) f(x), \tag{1}$$

where $\mu : \mathcal{X} \to \mathbb{R}_+$ is a weight function.

**Example 1** ($k$-means). For $\mathcal{X} \subset \mathbb{R}^d$ and $k \in \mathbb{N}$, the goal of $k$-means clustering is to find a set $\mathcal{C}^*$ of $k$ "cluster centers" by minimizing (1) over

$$\mathcal{F} = \left\{ f_{\mathcal{C}} : x \mapsto \min_{q \in \mathcal{C}} \|x - q\|_2^2 \ \mid \ \mathcal{C} \subset \mathbb{R}^d, |\mathcal{C}| = k \right\}.$$

Here, each query $f$ is indexed by a set of $k$ cluster centers, and the loss (1) is the quantization error.

**Example 2** (linear regression). When $\mathcal{X} = \{x_i := (y_i, z_i) \mid i \in [\![1, n]\!]\} \subset \mathbb{R}^{d+1}$, linear regression corresponds to minimizing (1) over

$$\mathcal{F} = \left\{ (y, z) \mapsto (a^\top y + b - z)^2 \mid a \in \mathbb{R}^d, b \in \mathbb{R} \right\}.$$

Penalty terms can be added to each function, to cover e.g. ridge or lasso regression.

In many machine learning applications, the complexity of the corresponding optimization problem grows with the cardinality $n$ of the dataset. When $n \gg 1$ makes optimization intractable, one is tempted to reduce the amount of data, using only a tractable number of representative samples. This is the idea formalized by *coresets*; we refer to (Bachem, Lucic, and Krause, 2017) for a survey, and to (Huang, Li, and Wu, 2024; Cohen-Addad, Larsen, Saulpic, Schwiegelshohn, and Sheikh-Omar, 2022) for specific coreset constructions for $k$-means and Euclidean clustering. An $\varepsilon$-coreset is a subset $\mathcal{S} \subset \mathcal{X}$, possibly with corresponding weights $\omega(x)$, $x \in \mathcal{S}$, such that

$$L_{\mathcal{S}}(f) := \sum_{x \in \mathcal{S}} \omega(x) f(x) \tag{2}$$

is within $\varepsilon$ of $L(f)$, uniformly in $f \in \mathcal{F}$. If the cardinality $m$ of $\mathcal{S}$ is significantly smaller than the intractable size $n$ of the original data set, one has reduced the complexity of the algorithm at a little cost in accuracy.

Many *randomized* coreset constructions, where such guarantees are shown to hold with large probability, are built by drawing elements *independently* from the data set $\mathcal{X}$ (Bachem et al., 2017, Chapter 3). Because a *representative* coreset should intuitively be made of *diverse* data points, *negative dependence* between the coreset elements has been proposed as an effective possibility to improve their performance (Tremblay, Barthelmé, and Amblard, 2019). In particular, the authors advocate the use of Determinantal Point Processes (DPPs), a family of probability distributions over subsets of $\mathcal{X}$ parametrized by an $n \times n$ kernel matrix $\mathbf{K}$ that enforces diversity, all of this while coming with a polynomial-time exact sampling algorithm.

Tremblay et al., 2019 give extensive theoretical and empirical justification for the use of DPPs in randomized coreset construction. In one of their key results, using concentration results in (Pemantle and Peres, 2011), Tremblay et al., 2019 bound the cardinality of a DPP-based $\varepsilon$-coreset, and their bound is $\mathcal{O}(\varepsilon^{-2})$. However, it is known that the best $\varepsilon$-coresets built with independent samples are also of cardinality $\mathcal{O}(\varepsilon^{-2})$. Thus, the crucial question of whether DPP-based coresets can provide a strict improvement remained to be settled; given the computational simplicity of independent schemes, this would be fundamental to justify the deployment of DPP-based methods.

In this paper, we settle this question in the affirmative, demonstrating that for carefully chosen kernels, DPP-based coresets provably yield significantly better accuracy guarantees than independent schemes; equivalently, to achieve similar accuracy it suffices to use significantly smaller coresets via DPPs. In particular, we will show that DPP-based coresets actually can achieve cardinality $m = \mathcal{O}(\varepsilon^{-2/(1+\delta)})$. The quantity $\delta$ depends on the variance of the subsampled loss under the considered DPP, and some DPPs yields $\delta > 0$. A cornerstone of our approach is a structural understanding of the coreset loss (2) as a so-called *linear statistic* of the random point set $\mathcal{S}$, which enables us to go beyond earlier results that were based on concentration properties of general Lipschitz functions of a DPP (Pemantle and Peres, 2011).

In this endeavour, we obtain very widely-applicable concentration inequalities for linear statistics of DPPs compared to the state of the art; cf. (Breuer and Duits, 2013) that mostly focuses on scalar-valued statistics for finite rank ensembles on $\mathbb{R}$. In particular, we are able to address all DPPs that

have appeared so far in the ML literature. Specifically, our results are able to handle *non-symmetric kernels* and *vector-valued* linear statistics.

DPPs with non-symmetric kernels have recently been shown to be of significant interest in machine learning, such as recommendation systems (Gartrell, Brunel, et al., 2019; Gartrell, Han, et al., 2020; Han et al., 2022), but they come with a limited theoretical toolbox, to which this paper makes a contribution. On the other hand, vector-valued statistics arise naturally in many learning problems, including coreset settings such as the gradient estimator in Stochastic Gradient Descent (Bardenet, Ghosh, et al., 2021). However, the literature on coresets for vector-valued statistics is scarce, and in this paper we inaugurate their study with effective approximation guarantees via DPPs.

The rest of the paper is organized as follows. Section 2 contains background on DPPs and coresets. Section 3 contains our contributions. Section 4 provides numerical illustrations. Section 5 contains a discussion on limitations and future work.

## 2   Background

We introduce here the two key notions of determinantal point process and coreset, and observe that a coreset guarantee is a uniform control over specific linear statistics of a point process.

**Determinantal point processes.** A point process $\mathcal{S}$ on a Polish space $\mathcal{X}$ is a random locally finite subset of $\mathcal{X}$. Given a reference measure $\mu$ on $\mathcal{X}$ (e.g., the Lebesgue measure if $\mathcal{X} = \mathbb{R}^d$ or the counting measure if $\mathcal{X}$ is discrete), a point process $\mathcal{S}$ is called a DPP (w.r.t. $\mu$) if there exists a measurable function $K : \mathcal{X} \times \mathcal{X} \to \mathbb{C}$ such that

$$\mathbb{E}\Big[ \sum_{\neq} f(x_{i_1}, \ldots, x_{i_k}) \Big] = \int_{\mathcal{X}^k} f(x_1, \ldots, x_k) \det[K(x_i, x_j)]_{k \times k} \, d\mu^{\otimes k}(x_1, \ldots, x_k), \qquad (3)$$

where the sum in the LHS ranges over all pairwise distinct $k$-tuples of the random locally finite subset $\mathcal{S}$, for all bounded measurable $f : \mathcal{X}^k \to \mathbb{R}$ and for all $k \in \mathbb{N}$. Such a function $K$ is called a *kernel* for the DPP $\mathcal{S}$, and $\mu$ is called the background measure.

When the ground set $\mathcal{X}$ is of finite cardinality $n$, an equivalent but more intuitive way to define DPPs is as follows: a random subset $\mathcal{S}$ of $\mathcal{X}$ is called a DPP if there exists an $n \times n$-matrix $\mathbf{K}$ such that

$$\mathbb{P}(T \subset \mathcal{S}) = \det[\mathbf{K}_T], \quad \forall T \subset \mathcal{X},$$

where $\mathbf{K}_T$ denotes the submatrix of $\mathbf{K}$ with rows and columns indexed by $T$.

In a similar vein to Gaussian processes, all the statistical properties of a DPP are encoded in this kernel function $K$ and background measure $\mu$. A feature of DPPs with far-reaching implications for machine learning is that sampling and inference with DPPs are tractable. We refer the reader to (Hough et al., 2006; Kulesza and Taskar, 2012) for general references. Originally introduced in electronic optics (Macchi, 1975), they have been turned into generic statistical models for repulsion in spatial statistics (Lavancier et al., 2014; Biscio and Lavancier, 2017) and machine learning (Kulesza and Taskar, 2012; Belhadji et al., 2020a; Brunel, 2018; Derezinski and Mahoney, 2019; Derezinski, Liang, et al., 2020; Gartrell, Brunel, et al., 2019; Ghosh and Rigollet, 2020).

**Example 3** (*L*-ensemble and m-DPP)**.** Let $\mathcal{X}$ be a finite set of cardinality $n$, $\mu$ be the counting measure, and $\mathbf{L}$ be a positive semi-definite $n \times n$-matrix. The $L$-ensemble with parameter $\mathbf{L}$ is the point process $\mathcal{S}$ on $\mathcal{X}$ such that, for all $T \subset \mathcal{X}$, $\mathbb{P}(\mathcal{S} = T) \propto \det[\mathbf{L}_T]$, where $\mathbf{L}_T$ is the square submatrix of $\mathbf{L}$ corresponding to the rows and columns indexed by the subset $T$. It can be shown that $\mathcal{S}$ is a DPP on $\mathcal{X}$ with kernel $\mathbf{K} := \mathbf{L}(\mathbf{I} + \mathbf{L})^{-1}$. In general, the cardinality of $\mathcal{S}$ is a random variable. By conditioning on the event $\{|\mathcal{S}| = m\}$, we obtain the so-called $m$-DPPs (Kulesza and Taskar, 2012).

**Example 4** (Multivariate OPE; Bardenet and Hardy, 2020)**.** Let $\mathcal{X} = \mathbb{R}^d$ and $\mu$ be a measure on $\mathbb{R}^d$ having all moments finite, let $(p_k)_{k \in \mathbb{N}^d}$ be the orthonormal sequence resulting from applying the Gram-Schmidt procedure to the monomials $x_1^{k_1} \ldots x_d^{k_d}$, taken in the graded lexical order. The kernel $K_\mu^{(m)}(x, y) := \sum_{k=0}^{m-1} p_k(x) p_k(y)$ then defines a projection DPP on $\mathbb{R}^d$, called the multivariate Orthogonal Polynomial Ensemble (OPE) of rank $m$ and reference measure $\mu$.

Multivariate OPEs were used in (Bardenet and Hardy, 2020) as nodes for numerical integration, leading to a Monte Carlo estimator with mean squared error decaying in $m^{-1-1/d}$, faster than under

independent sampling. In (Bardenet, Ghosh, and Lin, 2021), the authors investigated the problem of DPP-based minibatch sampling for Stochastic Gradient Descent (SGD), and exploited a delicate interplay between a finite dataset and its ambient data distribution to leverage this fast decay for improved approximation guarantees. In particular, they proposed the following DPP defined on a (large) finite ground set.

**Example 5** (Discretized multivariate OPE; Bardenet, Ghosh, et al., 2021). Let $n \in \mathbb{N}$ and $\mathcal{X} = \{x_1, \ldots, x_n\} \subset [-1, 1]^d$. Let $q(x)dx$ be a probability measure on $[-1, 1]^d$. Let $K_q^{(m)}$ be the multivariate OPE kernel of rank $m$ with reference measure $q(x)dx$, as defined in Example 4. Let $\tilde{\gamma} : [-1, 1]^d \to \mathbb{R}_+$ be a function, assumed to be positive on $\mathcal{X}$, and consider

$$K_{q,\tilde{\gamma}}^{(m)}(x, y) := \sqrt{\frac{q(x)}{\tilde{\gamma}(x)}} K_q^{(m)}(x, y) \sqrt{\frac{q(y)}{\tilde{\gamma}(y)}}, \quad x, y \in [-1, 1]^d.$$

Consider then the $n \times n$ matrix $\tilde{\mathbf{K}} = K_{q,\tilde{\gamma}}^{(m)}|_{\mathcal{X} \times \mathcal{X}}$. $\tilde{\mathbf{K}}$ is symmetric and positive semidefinite, and we let $\mathbf{K}$ be the matrix with the same eigenvectors, the $m$ largest eigenvalues replaced by 1, and the remaining eigenvalues replaced by 0. Then $\mathbf{K}$ defines a DPP on $\mathcal{X}$.

**Coresets.** Let $\varepsilon > 0$ and $\mathcal{X}$ be a set of cardinality $n$. The classical definition of a coreset is multiplicative.

**Definition 1** (multiplicative coreset). A subset[3] $\mathcal{S} \subset \mathcal{X}$ is an $\varepsilon$-multiplicative coreset if

$$\forall f \in \mathcal{F}, \ \left| \frac{L_\mathcal{S}(f)}{L(f)} - 1 \right| \leq \varepsilon, \tag{4}$$

where $L$ and $L_\mathcal{S}$ are respectively defined in (1) and (2).

An immediate and important consequence of (2) is that the ratio of the minimum value of $L_\mathcal{S}$ by that of $L$ is within $\mathcal{O}(\varepsilon)$ of 1 (Bachem et al., 2017, Theorem 2.1).

One way to satisfy (2) with high probability for a single $f$ is through importance sampling, taking $\mathcal{S}$ to be formed of $m > 0$ i.i.d. samples from some instrumental density $q$ on $\mathcal{X}$, and taking $\omega = \mu/q$ in (2). Langberg and Schulman, 2010 showed that a suitable choice of $q$ actually yields the uniform guarantee (2). It suffices to take for instrumental pdf $q(x) \propto \mu(x)s(x)$, where $s$ upper-bounds the so-called *sensitivity*

$$s(x) \geq \sup_{f \in \mathcal{F}} \frac{f(x)}{\sum_{y \in \mathcal{X}} \mu(y)f(y)}, \quad \forall x \in \mathcal{X}. \tag{5}$$

For $\delta > 0$, $k \geq \frac{S^2}{2\varepsilon^2} \log 2/\delta$ independent draws are then enough to build an $\varepsilon$-multiplicative coreset, where $S = \sum_{x \in \mathcal{X}} \mu(x)s(x)$; see (Bachem et al., 2017)[Section 2.3]. The tighter the bound (5), the smaller the size of the coreset. One important limitation is that finding a tight bound is nontrivial.

Although not standard, a natural alternative definition of a coreset is that of an additive coreset.

**Definition 2** (additive coreset). A subset $\mathcal{S} \subset \mathcal{X}$ is an $\varepsilon$-additive coreset if

$$\frac{1}{n} |L_\mathcal{S}(f) - L(f)| \leq \varepsilon, \quad \forall f \in \mathcal{F}. \tag{6}$$

Note the arbitrary scaling factor $1/n$ in (6) compared to (2), which we adopt to simplify comparisons between the two coreset definitions. With an additive coreset, the minimal value of $L_\mathcal{S}$ is guaranteed to be within $\pm n\varepsilon$ of the minimal value of $L$: Similarly to a multiplicative coreset, with $\varepsilon$ suitably small one should be happy to train one's algorithm only on $\mathcal{S}$.

**Coreset guarantee and linear statistics.** Let $\mathcal{S}$ be a point process on a finite $\mathcal{X} = \{x_1, \ldots, x_n\}$. For a test function $\varphi : \mathcal{X} \to \mathbb{R}$, we denote by $\Lambda(\varphi) := \sum_{x \in \mathcal{S}} \varphi(x)$ the so-called *linear statistic* of $\varphi$. In a coreset problem, for a query $f \in \mathcal{F}$, the estimated loss $L_\mathcal{S}(f)$ in (2) is the linear statistic $\Lambda(\omega f)$. When $\mathcal{S}$ is a DPP with a kernel $\mathbf{K}$ on $\mathcal{X}$ (w.r.t. the counting measure), we will choose the weight $\omega(x) = \mathbf{K}(x, x)^{-1}$, where for $x = x_i \in \mathcal{X}$, we define $\mathbf{K}(x, x)$ to be $\mathbf{K}_{ii}$. By (3), this choice makes $L_\mathcal{S}(f)$ an unbiased estimator for $L(f)$. Guaranteeing a coreset guarantee such as (6) with high probability thus corresponds to a uniform-in-$f$ concentration inequality for the linear statistic $\Lambda(\omega f)$. This motivates studying the concentration of linear statistics under a DPP, to which we now turn.

---

[3]Note that we defined a coreset as a subset and not a sub-multiset of $\mathcal{X}$, thus ignoring multiplicity. This is because we allow weights in (2), so that repeated items are unnecessary in a coreset.

# 3 Theoretical results

We first give new results on the concentration of linear statistics under very general DPPs. These results are of interest in their own right, and should find applications in ML beyond coresets. Next we examine the implications of the concentration of linear statistics for coresets, showing that a suitable DPP does yield a coreset size of size $o(\varepsilon^{-2})$, thus beating independent sampling.

**Concentration inequalities for linear statistics of DPPs.** We start with Hermitian kernels.

**Theorem 1** (Hermitian kernels). *Let $\mathcal{S}$ be a DPP on a Polish space $\mathcal{X}$ with reference measure $\mu$ and Hermitian kernel $K$. Then for any bounded test function $\varphi : \mathcal{X} \to \mathbb{R}$, we have*

$$\mathbb{P}(|\Lambda(\varphi) - \mathbb{E}[\Lambda(\varphi)]| \geq \varepsilon) \leq 2 \exp\Big(-\frac{\varepsilon^2}{4A \operatorname{Var}[\Lambda(\varphi)]}\Big), \quad \forall 0 \leq \varepsilon \leq \frac{2A \operatorname{Var}[\Lambda(\varphi)]}{3\|\varphi\|_\infty},$$

*where $A > 0$ is a universal constant.*

Our Theorem 1 is similar in spirit to a seminal concentration inequality by Breuer and Duits, 2013. However, their result only applies to DPPs with Hermitian projection kernels of finite rank. We emphasize that our Theorem 1 is applicable to all Hermitian kernels on general Polish spaces.

In view of recent interest in machine learning on DPPs with non-symmetric kernels, we present here a concentration inequality for such DPPs. We propose a novel approach to control the Laplace transform in the non-symmetric case (which can also be applied to the symmetric setting). As a trade-off, the range for $\varepsilon$ becomes a bit smaller. For simplicity, we present the result for a finite ground set, but the proof applies more generally.

**Theorem 2** (Non-symmetric kernels). *Let $\mathcal{S}$ be a DPP on a finite set $\mathcal{X} = \{x_1, \ldots, x_n\}$ with a non-symmetric kernel $\mathbf{K}$. Then for any bounded test function $\varphi : \mathcal{X} \to \mathbb{R}$, we have*

$$\mathbb{P}(|\Lambda(\varphi) - \mathbb{E}[\Lambda(\varphi)]| \geq \varepsilon) \leq 2 \exp\Big(-\frac{\varepsilon^2}{4 \operatorname{Var}[\Lambda(\varphi)]}\Big), \forall 0 \leq \varepsilon \leq \frac{\operatorname{Var}[\Lambda(\varphi)]^2}{40\|\varphi\|_\infty^3 \cdot \max(1, \|\mathbf{K}\|_{\mathrm{op}}^2) \cdot \|\mathbf{K}\|_*},$$

*where $\|\cdot\|_{\mathrm{op}}$ denotes the spectral norm and $\|\cdot\|_*$ denotes the nuclear norm of a matrix.*

**Remark 2.1.** *For simplicity, we will use the concentration inequality in Theorem 1 from now on. However, we keep in mind that we always can apply Theorem 2 to deduce analogous results for non-symmetric kernels.*

We conclude with a concentration inequality for linear statistics of vector-valued functions.

**Theorem 3** (Vector-valued statistics). *Let $\mathcal{S}$ be a DPP on a Polish space $\mathcal{X}$ with reference measure $\mu$ and Hermitian kernel $K$. Let $\Phi = (\varphi_1, \ldots, \varphi_p)^\top : \mathcal{X} \to \mathbb{R}^p$ be a vector-valued test function, and we denote by $\Lambda(\Phi)$, $\mathbb{V}(\Phi)$ the vectors $(\Lambda(\varphi_i))_{i=1}^p$ and $(\operatorname{Var}[\Lambda(\varphi_i)]^{1/2})_{i=1}^p$, respectively. Let $\|x\|_\omega^2 := \sum_{i=1}^p \omega_i^2 |x_i|^2$ be a weighted norm on $\mathbb{R}^p$ for some weights $\omega_1, \ldots \omega_p \geq 0$. Then, for some universal constant $A > 0$, we have*

$$\mathbb{P}(\|\Lambda(\Phi) - \mathbb{E}[\Lambda(\Phi)]\|_\omega \geq \varepsilon) \leq 2p \exp\Big(-\frac{\varepsilon^2}{4A\|\mathbb{V}(\Phi)\|_\omega^2}\Big),$$

*for $0 \leq \varepsilon \leq \frac{2A\|\mathbb{V}(\Phi)\|_\omega}{3} \min_{1 \leq i \leq p} \frac{\sqrt{\operatorname{Var}[\Lambda(\varphi_i)]}}{\|\varphi_i\|_\infty}$.*

**DPPs for coresets.** We demonstrate the effectiveness of concentration inequalities for linear statistics of DPPs in the coresets problem, achieving uniform approximation guarantees over function classes. To accommodate as many ML settings as possible, we shall consider two natural types of function classes: vector spaces of functions (A.1) and parametrized function spaces (A.2).

For vector spaces of functions, we assume that

$$\dim \operatorname{span}_{\mathbb{R}}(\mathcal{F}) = D < \infty \text{ for some } D. \tag{A.1}$$

This assumption covers common situations like linear regression in Example 2, where we observe that each $f \in \mathcal{F}$ is a quadratic function in $(d+1)$ variables. Thus the dimension of the linear span of $\mathcal{F}$ is at most $(d+1)^2 + (d+1) + 1$. Another popular class of queries, originating in signal processing

problems, is the class of *band-limited functions*. A function $f : \mathbb{T}^d \mapsto \mathbb{R}$ (where $\mathbb{T}^d$ denotes the $d$-dimensional torus) is said to be *band-limited* if there exists $B \in \mathbb{N}$ such that its Fourier coefficients $\hat{f}(k_1, \ldots, k_d) = 0$ whenever there is a $k_j$ such that $|k_j| > B$. It is easy to see that the space $\mathcal{F}$ of $B$-bandlimited functions satisfies $\dim \mathcal{F} \leq (2B + 1)^d$.

Another common scenario is when $\mathcal{F}$ is parametrized by a finite-dimensional parameter space:

$$\mathcal{F} = \{f_\theta : \theta \in \Theta\}, \text{ where } \Theta \text{ is a bounded subset of } \mathbb{R}^D \text{ for some } D, \tag{A.2}$$

$$\|f_\theta - f_{\theta'}\|_\infty \leq \ell \|\theta - \theta'\| \text{ for some } \ell > 0, \text{ uniformly on } \Theta. \tag{A.3}$$

Conditions (A.2) and (A.3) cover e.g. the $k$-means problem of Example 1, as well as (non-)linear regression settings. For $k$-means, for instance, each query is parametrized by its cluster centers $\mathcal{C} = \{q_1, \ldots, q_k\}$, which can be viewed as a parameter $(q_1, \ldots, q_k) \in \mathbb{R}^{kd}$.

Finally, with the idea in mind to derive multiplicative coresets from additive ones, we note that since $L(f)$ is typically of order $n$ (for any $f$ whose effective support covers a positive fraction of the ground set), it is natural to assume that

$$\frac{1}{n}|L(f)| \geq c, \text{ for some } c > 0, \text{ uniformly on } \mathcal{F}. \tag{A.4}$$

**Theorem 4.** *Let $\mathcal{S}$ be a DPP with a Hermitian kernel $\mathbf{K}$ on a finite set $\mathcal{X} = \{x_1, \ldots, x_n\}$ and $m = \mathbb{E}[|\mathcal{S}|]$. Assume that for all $i \in \{1, \ldots, n\}$, $\mathbf{K}_{ii} \geq \rho \cdot m/n$ for some $\rho > 0$ not depending on $m, n$. Let $V \geq \sup_{f \in \mathcal{F}} \mathbb{V}\mathrm{ar}\left[n^{-1} L_\mathcal{S}(f)\right]$. Under (A.1) and (A.4),*

$$\mathbb{P}\left(\exists f \in \mathcal{F} : \left|\frac{L_\mathcal{S}(f)}{L(f)} - 1\right| \geq \varepsilon\right) \leq 2\exp\left(6D - \frac{c^2 \varepsilon^2}{16 A V}\right), \quad 0 \leq \varepsilon \leq \frac{4 A \rho m V}{3 c \sup_{f \in \mathcal{F}} \|f\|_\infty}.$$

*Assuming (A.2), (A.3), (A.4) and $|\mathcal{S}| \leq B \cdot m$ a.s. for some $B > 0$, we have*

$$\mathbb{P}\left(\exists f \in \mathcal{F} : \left|\frac{L_\mathcal{S}(f)}{L(f)} - 1\right| \geq \varepsilon\right) \leq 2\exp\left(CD - D\log \varepsilon - \frac{c^2 \varepsilon^2}{16 A V}\right), \ 0 \leq \varepsilon \leq \frac{4 A \rho m V}{3 c \sup_{f \in \mathcal{F}} \|f\|_\infty}.$$

*Here $A > 0$ is a universal constant and $C = C(\Theta, B, \rho, \ell, c) > 0$ is some constant.*

**Remark 4.1.** *For a bounded query $f$, $\mathbb{V}\mathrm{ar}\left[n^{-1} L_\mathcal{S}(f)\right] = \mathcal{O}(m^{-1})$ for i.i.d. sampling. In comparison, sampling with DPPs often yields smaller variance for linear statistics, in $\mathcal{O}(m^{-(1+\delta)})$ for some $\delta > 0$; see Section 3 for an example. Thus, the upper bound for the range of $\varepsilon$ for which we could use our concentration result is $\mathcal{O}(m^{-\delta})$. Plugging in $\varepsilon = m^{-\alpha}$ for $\alpha \geq \delta$ gives the upper bounds $2\exp(6D - C'm^{1+\delta-2\alpha})$ and $2\exp(CD + \alpha D \log m - C'm^{1+\delta-2\alpha})$ respectively ($C$ and $C'$ are some positive constants independent of $m$ and $n$), which both converge to 0 as $m \to \infty$ as long as $\alpha < (1 + \delta)/2$. In other words, the accuracy rate $\varepsilon$ can be chosen to be as small as $m^{-1/2-\delta'/2}$, for any $0 < \delta' < \delta$, which is strictly smaller than the best accuracy rate $m^{-1/2}$ of i.i.d. sampling.*

**Remark 4.2.** *For i.i.d. sampling $\mathcal{S}$ with expected size $m$, $\mathbb{P}(x \in \mathcal{S}) = m/n$ for all $x \in \mathcal{X}$. For a DPP $\mathcal{S}$ with kernel $\mathbf{K}$, one has $\mathbb{P}(x_i \in \mathcal{S}) = \mathbf{K}_{ii}$. Thus, assuming that for all $i$, $\mathbf{K}_{ii} \geq \rho \cdot m/n$ for some $\rho > 0$ means that every point in the dataset $\mathcal{X}$ should have a reasonable chance to be sampled. This also guarantees that the estimated loss $L_\mathcal{S}(f) = \sum_{x \in \mathcal{S}} f(x)/\mathbf{K}(x, x)$ will not blow up, where for $x = x_i \in \mathcal{X}$, we write $\mathbf{K}(x, x)$ for $\mathbf{K}_{ii}$.*

**Remark 4.3.** *For the parametrized function spaces, the assumption $|\mathcal{S}| \leq B \cdot m$ a.s. is not strictly necessary, and is introduced here only for the sake of simplicity in presenting the results. A version of Theorem 4 without this assumption will be discussed in Appendix A.4. In fact, we only need $n^{-1} \sum_{x \in \mathcal{S}} \mathbf{K}(x, x)^{-1}$ to be bounded with high probability, which follows from the condition $\mathbf{K}(x, x) \geq \rho \cdot m/n$ and the fact that $|\mathcal{S}|$ is highly concentrated around its mean $m$.*

**Remark 4.4.** *However, we remark that the assumption $|\mathcal{S}| \leq B \cdot m$ a.s. holds for most kernels of interest; DPPs with projection kernels being typical and significant examples. In machine learning terms, it entails that the coresets are not much bigger than their expected size $m$; whereas in practice, sampling schemes typically produce coresets of a fixed size (such as with projection DPPs).*

**Remark 4.5.** *It is straightforward to derive a version for additive coresets from Theorem 1. In fact, we will not need assumption (A.4) in the additive setting.*

For the coresets problem for vector-valued functions, let $\mathcal{F}$ consist of $\mathbf{f} : \mathcal{X} \to \mathbb{R}^p$. For each $\mathbf{f} \in \mathcal{F}$, we denote by $L_{\mathcal{S}}(\mathbf{f}), L(\mathbf{f})$ and $\mathbb{V}(\mathbf{f})$ the vectors in $\mathbb{R}^p$ whose $i$-coordinates are $L_{\mathcal{S}}(f_i), L(f_i)$ and $\mathbb{V}\mathrm{ar}\left[n^{-1}L_{\mathcal{S}}(f_i)\right]^{1/2}$, respectively. Let $\|x\|_\omega^2 := \sum_{i=1}^p \omega_i^2 |x_i|^2$ be a weighted norm on $\mathbb{R}^p$.

**Theorem 5.** *Let $\mathcal{S}$ be a DPP as in Theorem 4. Let $V \geq \sup_{\mathbf{f} \in \mathcal{F}} \max_{1 \leq i \leq p} \omega_i^2 \, \mathbb{V}\mathrm{ar}\left[n^{-1}L_{\mathcal{S}}(f_i)\right]$. Assuming* (A.1), *then*

$$\mathbb{P}\Big(\exists \, \mathbf{f} \in \mathcal{F} : \frac{1}{n}\|L_{\mathcal{S}}(\mathbf{f}) - L(\mathbf{f})\|_\omega \geq \varepsilon\Big) \leq 2p \exp\Big(6D - \frac{c^2 \varepsilon^2}{16AV}\Big),$$

*where $0 \leq \varepsilon \leq \frac{4A\rho m V}{3c} \cdot (\sup_{\mathbf{f} \in \mathcal{F}} \max_{i=1,\dots,p} \|f_i\|_\infty)^{-1}$.*

**Application: Discretized multivariate OPE.**   We revisit Example 5. It has been shown in (Bardenet, Ghosh, et al., 2021) that sampling with the DPP $\mathcal{S}$ constructed in this example yields significant variance reduction for a wide class of linear statistics on $\mathcal{X}$. To be more precise, in their setting, $\mathcal{X} = \{x_1, \dots, x_n\}$ is a random data set, where $x_i$'s are i.i.d. samples from a distribution $\gamma$ with support inside a $d$-dimensional hypercube; $\tilde{\gamma}$ is a density estimator for $\gamma$ and $q(x)dx$ is a reference measure on that hypercube. The DPP $\mathcal{S}$ is then defined by the kernel $\mathbf{K}$ in Example 5 w.r.t. the empirical measure $n^{-1} \sum_{x \in \mathcal{X}} \delta_x$. We normalize the kernel by setting $\hat{\mathbf{K}} := n^{-1}\mathbf{K}$, so that $\mathcal{S}$ is a DPP with kernel $\hat{\mathbf{K}}$ w.r.t. the counting measure on $\mathcal{X}$. Then, under some mild assumptions on $\tilde{\gamma}$ and $q(x)dx$, with high probability in the data set $\mathcal{X}$, we have $\mathbb{V}\mathrm{ar}\left[n^{-1}L_{\mathcal{S}}(f)\right] = \mathcal{O}(m^{-(1+1/d)})$ for any test function $f$ satisfying some mild regularity conditions. For more details, we refer the reader to (Bardenet, Ghosh, et al., 2021).

The significant reduction on the variance of linear statistics motivates us to apply Theorem 4 to this setting. We also remark that all assumptions on the kernel in Theorem 4 are satisfied for $\hat{\mathbf{K}}$ with high probability in the data set $\mathcal{X}$ (see Appendix A.6). Let $\mathcal{F}$ be a family of test functions on $\mathcal{X}$ satisfying regularity conditions as in Bardenet, Ghosh, et al., 2021. Then we can state:

**Theorem 6.** *For $\varepsilon = \mathcal{O}(m^{-1/d})$, w.h.p. in the data set $\mathcal{X}$, we have*

$$\mathbb{P}_{\mathcal{S}}\Big(\exists f \in \mathcal{F} : \Big|\frac{L_{\mathcal{S}}(f)}{L(f)} - 1\Big| \geq \varepsilon\Big) \leq 2\exp\Big(6D - C'\varepsilon^2 m^{1+1/d}\Big), \quad \text{assuming (A.1)} \,,$$

*and*

$$\mathbb{P}_{\mathcal{S}}\Big(\exists f \in \mathcal{F} : \Big|\frac{L_{\mathcal{S}}(f)}{L(f)} - 1\Big| \geq \varepsilon\Big) \leq 2\exp\Big(CD - D\log\varepsilon - C'\varepsilon^2 m^{1+1/d}\Big), \;\; \text{assuming (A.2), (A.3),}$$

*where $\mathbb{P}_{\mathcal{S}}$ indicates the randomness only in $\mathcal{S}$ and $C, C' > 0$ do not depend on $m, n$.*

**Remark 6.1.** *Theorem 6 confirms the discussion in Remark 4.1 for this particular example of DPP. More precisely, Theorem 6 implies that, with probability tending to 1, sampling with this DPP gives $|L_{\mathcal{S}}(f)/L(f) - 1| \leq m^{-(\frac{1}{2}+\frac{1}{2d})^-}$, $\forall f \in \mathcal{F}$, where $(\frac{1}{2} + \frac{1}{2d})^-$ denotes any positive number strictly smaller than $\frac{1}{2} + \frac{1}{2d}$. Meanwhile, for i.i.d. sampling, the accuracy rate $\varepsilon$ is at best $m^{-1/2}$.*

**Remark 6.2.** *It may be noted that, DPPs being Hilbert space-based models, they interact well with linear projection based methods. As such, our method can be applied on dimensionally reduced data, wherein the $d$ in Remark 6.1 can be taken to be the reduced dimension, which is usually quite small. As such, the improvement in the approximation guarantees is substantial, especially for large scale problems entailing large $m$.*

## 4   Experiments

In this section, we compare randomized coresets for the $k$-means problem of Example 1, on different datasets. One virtue of $k$-means as a benchmark is that asymptotically tight upper-bound on sensitivity (5) can easily be computed (Bachem et al., 2017, Lemma 2.2).

**Competing approaches.**   We compare 6 different coreset samplers.[4] Each sampler takes as input a finite dataset $\mathcal{X}$, an integer $m$, and sampler-specific parameters. It returns a random subset $\mathcal{S} \subset \mathcal{X}$ of

---

[4]The link to a GitHub repository is temporarily hidden for anonymity.

cardinality $m$. For the associated weight function $\omega$ in (2), we always take the inverse of the marginal probability of inclusion, i.e. $\omega(x) = 1/\mathbb{P}(x \in \mathcal{S})$.

The first two baselines use independent sampling. The `uniform` method returns $m$ samples from $\mathcal{X}$, uniformly and without replacement, and runs in $\mathcal{O}(m)$. The second method, `sensitivity`, is specific to the $k$-means problem. It corresponds to the classical sensitivity-based importance sampling coreset of Langberg and Schulman, 2010 described in Section 2. It runs in $\mathcal{O}(nk + nm)$.

The rest of the methods use negative dependence. The third method, termed `G-mDPP`, uses an $m$-DPP sampler where the likelihood kernel is a Gaussian kernel, with adjustable bandwidth denoted by $h$. It is basically Algorithm 1 of Tremblay et al., 2019, except we do not approximate the likelihood kernel using random features. We prefer avoiding approximations in this paper to isolatedly probe the benefit of negative dependence, but our choice comes at the cost $\mathcal{O}(n^3)$ of performing SVD as a preprocessing, in addition to the usual $\mathcal{O}(nm^2)$ sampling time. Similarly, we compute the marginal probabilities of inclusion of $m$-DPPs exactly, via Equation (205) and Algorithm 7 of Kulesza and Taskar, 2012. These costly steps will likely be approximated in real data applications; see the discussion of complexity to Section 5. The fourth method, `OPE`, is the discretized OPE of Example 5. We take $q$ to be a product of univariate beta pdfs, with parameters tuned to match the marginal moments of the dataset, as in (Bardenet, Ghosh, et al., 2021). We take $\tilde{\gamma}$ to be a kernel density estimator (KDE) built on $\mathcal{X}$, using the Epanechnikov kernel, with Scott's bandwidth selection method, as implemented in the `scikit-learn` package (Pedregosa et al., 2011). When KDE estimation is precomputed as in our experiments, the method runs in $\mathcal{O}(nm^2)$, and $\mathcal{O}(n^2 + nm^2)$ otherwise. Note that there is no cubic power of $n$, as one can perform the eigenvalue thresholding in Example 1 by a reduced SVD of the $m \times n$ feature matrix $(p_k(x_i))$. The fifth method, termed `Vdm-DPP`, is Algorithm 2 of Tremblay et al., 2019, which runs in $\mathcal{O}(nm^2)$. It is an OPE in the sense of Example 4, but where the reference measure $\mu$ is the discrete empirical measure of the dataset. Although we have no result on how its linear statistics scale, its similarity with the discretized OPE, as well as its numerical performance in the experiments of Tremblay et al., 2019, make us expect `Vdm-DPP` to behave similarly to `OPE`. The sixth method, `stratified`, is a stratified sampling baseline limited to the case where $\mathcal{X} \subset [-1, 1]^d$ and $\mathcal{X}$ is "well-spread". It partitions $[-1, 1]^d$ into a grid of $m$ bins, and then independently draws one element uniformly in the intersection of $\mathcal{X}$ with each bin. It is a special case of projection DPP, which runs in $\mathcal{O}(nm)$ and has obvious pitfalls, like requiring that $\mathcal{X}$ has a non-empty intersection with each bin, which is unlikely to be the case for non-uniformly spread datasets and high dimensions. Yet, this is a simple solution that one would likely implement to probe the benefits of negative dependence.

**The performance metric.** To investigate the cardinality of a coreset for a given error, we let $Q_{\mathcal{S}}$ denote the quantile function of $\sup |L_{\mathcal{S}}(f) - L(f)|/L(f)$, the supremum over all queries of the relative error. Intuitively, $Q_{\mathcal{S}}(0.9) = 10^{-2}$ means that 90% of the sampled coresets have a worst case relative error below $10^{-2}$. We shall look at how an estimated $Q_{\mathcal{S}}(0.9)$ varies with $m$, especially its slope in log-log plots with respect to $m$. Now, the set $\mathcal{F}$ of all queries for $k$-means in combinatorially large, even for small values of $k$. Therefore, each time we need to evaluate the supremum of the relative error, we rather uniformly sample without replacement $k$ elements of $\mathcal{X}$, 100 times and independently, and we take the maximum value of the relative error among these 100 values. Moreover, for each method and each coreset size $m$, the quantile function $Q_{\mathcal{S}}(0.9)$ is estimated by an empirical quantile over 100 independent coresets sampled for each value of $m$.

**Results.** We first consider a synthetic dataset of $n = 1024$ data points, sampled uniformly and independently in $[-1, 1]^d$; see Figure 1a. We consider $d = 2$ for demonstration purposes, but we have observed similar results for other small dimensions. Figure 1b depicts our estimate of $Q_{\mathcal{S}}(0.9)$ as a function of the coreset size $m$, in log-log format. The two i.i.d. baselines decrease as $m^{-1/2}$, as expected. The stratified baseline, intuitively well-suited to uniformly-spread datasets, outperforms all other methods with a $m^{-1}$ rate, consistent with its known optimal variance reduction (Novak, 1988). Finally, the $m$-DPP and the two DPPs also yield a faster decay, eventually outperforming the i.i.d. baselines as $m$ grows. This is expected for the discretized OPE, as it follows from the theoretical results from Section 3; but it is interesting to see that the Gaussian $m$-DPP and the `Vdm-DPP` seem to reach a similar $m^{-3/4}$ fast rate. For the Gaussian $m$-DPP, however, the performance depends on the value of the bandwidth of the Gaussian kernel: in Figure 1c, we see that the rate of decay can go from i.i.d.-like to OPE-like as the bandwidth increases; this is expected from results like (Barthelmé et al., 2023). Note that the color code of Figure 1c differs from other figures.

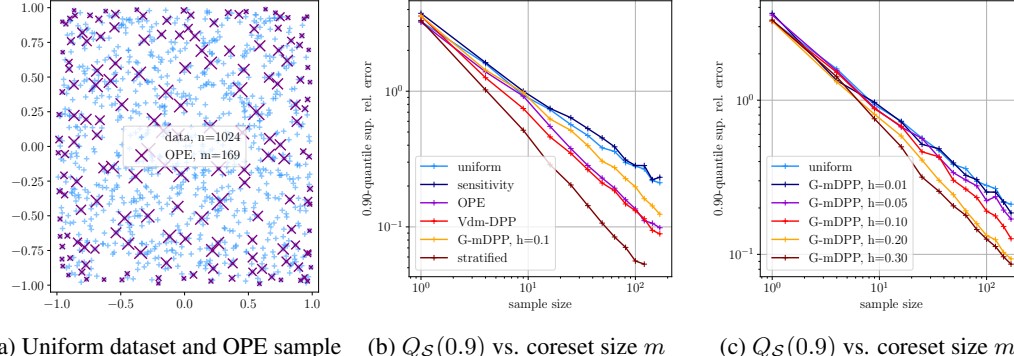

(a) Uniform dataset and OPE sample   (b) $Q_{\mathcal{S}}(0.9)$ vs. coreset size $m$   (c) $Q_{\mathcal{S}}(0.9)$ vs. coreset size $m$

Figure 1: Results for the uniform dataset.

In the uniform dataset of Fig. 1a, the sensitivity function is almost flat, which makes `sensitivity` behave like `uniform`. To give an edge to `sensitivity`, we now consider the trimodal dataset shown in Fig. 2a, with an OPE sample superimposed. The performance of `sensitivity` improves; see Figure 2b, while the determinantal samplers still outperform the independent ones thanks to a faster decay. For this dataset, it is not easy to stratify, and we thus do not show results for `stratified`. We note that the size of a marker placed at $x$ is proportional to the corresponding weight $1/K(x, x)$ in the estimator of the average loss. Equivalently, the marker size is inversely proportional to the marginal probability of $x$ being included in the DPP sample.

Finally, we consider the classical MNIST dataset, after a PCA of dimension 4. Figure 2c shows again the faster decay of the performance metric for the two DPPs (`OPE` and `Vdm-DPP`), compared to the two independent methods. However, the advantage progressively disappears as the dimension increases beyond 4 (unshown), as expected from the gain in variance of the discretized multivariate OPE, which becomes negligible when $d \gg 1$; see Section 5 for suggestions on how to prove a dimension-independent decay. The source code used in this work is available at github.com/hsimonfroy/DPPcoresets, where DPP samplers are built upon the Python package DPPy (Gautier et al., 2019).

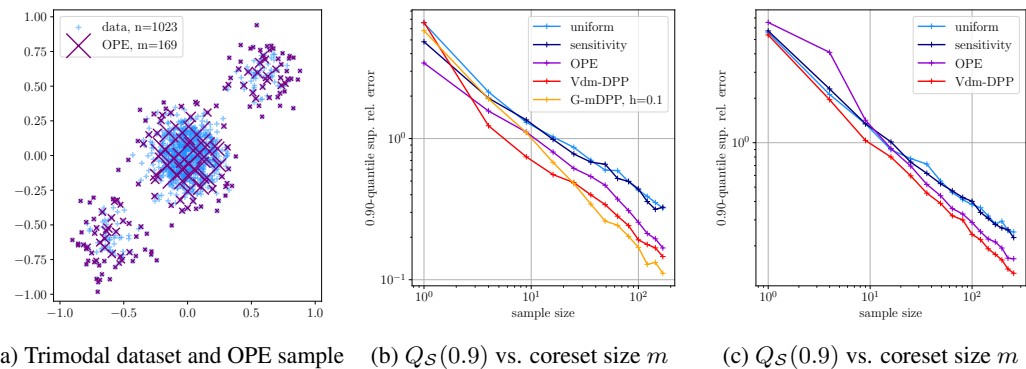

(a) Trimodal dataset and OPE sample   (b) $Q_{\mathcal{S}}(0.9)$ vs. coreset size $m$   (c) $Q_{\mathcal{S}}(0.9)$ vs. coreset size $m$

Figure 2: Results on other datasets.

## 5   Discussion

**Limitations.**   Our paper is a theoretical contribution, and our approach has several limitations before it can be a *practical* addition to the coreset toolbox. The improvement over independent sampling relies on a variance scaling for linear statistics of a particular DPP, which itself relies on both 1) an Ansatz that the dataset was generated i.i.d. from some pdf $\gamma$ with a large support, and 2) the availability of a good approximation to $\gamma$; see Section 3 and (Bardenet, Ghosh, et al., 2021). While Item 1) is usually deemed to be reasonable in a wide range of situations, we solve Item 2) by

relying on a kernel density estimator, which is costly to manipulate. Another limitation is that the improvement over independent sampling is in $1/d$ and thus progressively vanishes as the dimension increases. Finally, a classical caveat is that although tractable, sampling a DPP still costs $\mathcal{O}(nm^2)$, provided the kernel is available in diagonalized form.

**Future work.** The limitations above set up a research program. In particular, an intriguing observation in our empirical studies is the comparative performance of various DPP-based coreset samplers; several of them exhibit effective performance. While we have sharp theoretical guarantees for the discretized OPE-based scheme, obtaining similar guarantees and parameter-tuning protocols for other samplers, like $m$-DPPs, will be of great practical interest as they would bypass the need, e.g., for an approximation to the data-generating mechanism $\gamma$. The DPP called Vdm-DPP in Section 4, which is itself an OPE for a discrete measure, might be a bridge between OPEs and $m$-DPPs, as Vdm-DPP can be seen as a limit of Gaussian $m$-DPPs (Barthelmé et al., 2023). On a more general note, improving the computational complexity of sampling DPPs remains an active topic, and we should examine which techniques, e.g. by leveraging low-rank structures, preserve the small coreset property. Any breakthrough in the complexity of DPP sampling would also have salutary consequences for the broader program of negative dependence as a toolbox for machine learning. On the dependence of the rate to the dimension, we propose to investigate the impact of smoothness of the test functions on the rate: in numerical integration with mixtures of DPPs, smoothness does bring dimension-independent rates (Belhadji et al., 2020b). Finally, in a more theoretical direction, extending concentration inequalities for linear statistics beyond the restricted range of $\varepsilon$ appearing e.g. in Theorem 1 is a mathematically challenging problem, with potential learning-theoretic consequences.

## Acknowledgments and Disclosure of Funding

RB and HSO acknowledge support from ERC grant Blackjack (ERC-2019-STG-851866) and ANR AI chair Baccarat (ANR-20-CHIA-0002). SG was supported in part by the MOE grants R-146-000-250-133, R-146-000-312-114, A-8002014-00-00 and MOE-T2EP20121-0013. HST was supported by the NUS Research Scholarship.

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

# A    Appendix / supplemental material

## A.1    Proof of Theorem 1

We present here the proof for the general setting, i.e., when $\mathcal{X}$ is a Polish space and $\mathcal{S}$ is a DPP with a Hermitian kernel $K$ w.r.t a background measure $\mu$. By abuse of notation, we will also denote by $K$ the integral operator

$$K : L^2(\mathcal{X}, \mu) \to L^2(\mathcal{X}, \mu) \quad , \quad f(x) \mapsto \int_{\mathcal{X}} K(x, y) f(y) \mathrm{d}\mu(y).$$

We denote by $C_k(\varphi), k \geq 1$ the cumulants of $\Lambda(\varphi)$, i.e.

$$\log \mathbb{E}[e^{t\Lambda(\varphi)}] = \sum_{k \geq 1} \frac{C_k(\varphi)}{k!} t^k, \quad \text{for } t \text{ near } 0.$$

Note that $C_1(\varphi) = \mathbb{E}[\Lambda(\varphi)], C_2(\varphi) = \mathbb{V}\mathrm{ar}\,[\Lambda(\varphi)]$. In general, we have the formula (see Johansson and Lambert, 2018)

$$C_k(\varphi) = \sum_{q=1}^{k} \frac{(-1)^{q+1}}{q} \sum_{\substack{k_1, \ldots, k_q \geq 1 \\ k_1 + \ldots + k_q = k}} \frac{k!}{k_1! \ldots k_q!} \mathrm{Tr}[\Phi^{k_1} K \ldots \Phi^{k_q} K], \tag{7}$$

where $\Phi : L^2(\mathcal{X}, \mu) \to L^2(\mathcal{X}, \mu)$ is the operator $f(x) \mapsto \varphi(x) f(x)$.

By the Macchi-Soshnikov theorem (Macchi, 1972; Soshnikov, 2000), $0 \preceq K \preceq I$, and we can write

$$C_2(\varphi) = \mathrm{Tr}[\Phi(I - K)\Phi K] = \mathrm{Tr}[\sqrt{K}\Phi(I - K)\Phi\sqrt{K}] = \|\sqrt{I - K}\Phi\sqrt{K}\|_{\mathrm{HS}}^2,$$

where $\|\cdot\|_{\mathrm{HS}}$ denotes the Hilbert-Schmidt norm of an operator.

**Lemma 1.** *For $k \geq 1$, we have*

$$\|\sqrt{I - K}\Phi^k\sqrt{K}\|_{\mathrm{HS}} \leq k\|\varphi\|_\infty^{k-1}\|\sqrt{I - K}\Phi\sqrt{K}\|_{\mathrm{HS}},$$

*where $\|\varphi\|_\infty := \sup_{x \in \mathcal{X}} |\varphi(x)|$.*

*Proof.* One has

$$
\begin{aligned}
\|\sqrt{I - K}\Phi^k\sqrt{K}\|_{\mathrm{HS}}^2 &= \mathrm{Tr}[\sqrt{K}\Phi^k(I - K)\Phi^k\sqrt{K}] \\
&= \mathrm{Tr}[\Phi^k(I - K)\Phi^k K] \\
&= \mathrm{Tr}[\Phi^{2k} K] - \mathrm{Tr}[\Phi^k K \Phi^k K] \\
&= \int \varphi(x)^{2k} K(x, x) \mathrm{d}\mu(x) - \iint \varphi(x)^k K(x, y) \varphi(y)^k K(y, x) \mathrm{d}\mu(x) \mathrm{d}\mu(y) \\
&= \int \varphi(x)^{2k} \Big( K(x, x) - \int K(x, y) K(y, x) \mathrm{d}\mu(y) \Big) \mathrm{d}\mu(x) \\
&\quad + \frac{1}{2} \iint (\varphi(x)^k - \varphi(y)^k)^2 K(x, y) K(y, x) \mathrm{d}\mu(x) \mathrm{d}\mu(y).
\end{aligned}
$$

Since $0 \preceq K \preceq I$, we have $K^2 \preceq K$, which implies $K(x, x) \geq \int K(x, y) K(y, x) \mathrm{d}\mu(y)$ for $\mu$-a.e. $x$. Thus,

$$
\begin{aligned}
&\int \varphi(x)^{2k} \Big( K(x, x) - \int K(x, y) K(y, x) \mathrm{d}\mu(y) \Big) \mathrm{d}\mu(x) \\
&\leq \quad \|\varphi\|_\infty^{2k-2} \int \varphi(x)^2 \Big( K(x, x) - \int K(x, y) K(y, x) \mathrm{d}\mu(y) \Big) d\mu(x).
\end{aligned}
$$

On the other hand, by the symmetry of $K$, we have $K(x, y) K(y, x) = |K(x, y)|^2 \geq 0$ for all $x, y \in \mathcal{X}$. Note that

$$|\varphi(x)^k - \varphi(y)^k| = |\varphi(x) - \varphi(y)| \Big| \sum_{j=0}^{k-1} \varphi(x)^j \varphi(y)^{k-1-j} \Big| \leq k\|\varphi\|_\infty^{k-1} |\varphi(x) - \varphi(y)|.$$

Combining all ingredients, we deduce that $\|\sqrt{I - K}\Phi^k\sqrt{K}\|_{\mathrm{HS}}^2 \leq k^2\|\varphi\|_\infty^{2k-2}\|\sqrt{I - K}\Phi\sqrt{K}\|_{\mathrm{HS}}^2$, as desired. $\qquad\square$

**Lemma 2.** *For $k \geq 3$, we have*

$$\frac{|C_k(\varphi)|}{k!} \leq \frac{1}{\sqrt{2\pi}} e^k k^{3/2} \|\varphi\|_\infty^{k-2} C_2(\varphi).$$

*Proof.* We recall the formula (7), observe that

$$\sum_{q=1}^{k} \frac{(-1)^{q+1}}{q} \sum_{\substack{k_1,\ldots,k_q \geq 1 \\ k_1+\ldots+k_q=k}} \frac{k!}{k_1! \ldots k_q!} = 0.$$

Then one can write

$$
\begin{aligned}
C_k(\varphi) &= \sum_{q=1}^{n} \frac{(-1)^{q+1}}{q} \sum_{\substack{k_1,\ldots,k_q \geq 1 \\ k_1+\ldots+k_q=k}} \frac{k!}{k_1! \ldots k_q!} \Big( \mathrm{Tr}[\Phi^{k_1} K \ldots \Phi^{k_q} K] - \mathrm{Tr}[\Phi^k K] \Big) \\
&= \sum_{q=2}^{n} \frac{(-1)^{q+1}}{q} \sum_{\substack{k_1,\ldots,k_q \geq 1 \\ k_1+\ldots+k_q=k}} \frac{k!}{k_1! \ldots k_q!} \Big( \mathrm{Tr}[\Phi^{k_1} K \ldots \Phi^{k_q} K] - \mathrm{Tr}[\Phi^k K] \Big).
\end{aligned}
$$

For any $k_1, \ldots, k_q \geq 1$ such that $k_1 + \ldots + k_q = k$, we observe that

$$
\begin{aligned}
&|\mathrm{Tr}[\Phi^{k_1} K \ldots \Phi^{k_{q-2}} K \Phi^{k_{q-1}+k_q} K] - \mathrm{Tr}[\Phi^{k_1} K \ldots \Phi^{k_{q-2}} K \Phi^{k_{q-1}} K \Phi^{k_q} K]| \\
&= |\mathrm{Tr}[\Phi^{k_1} K \ldots \Phi^{k_{q-2}} K \Phi^{k_{q-1}} (I - K) \Phi^{k_q} K]| \\
&= |\mathrm{Tr}[\sqrt{K} \Phi^{k_1} K \ldots \Phi^{k_{q-2}} \sqrt{K} \sqrt{K} \Phi^{k_{q-1}} \sqrt{I-K} \sqrt{I-K} \Phi^{k_q} \sqrt{K}]| \\
&\leq \|\sqrt{K} \Phi^{k_1} K \ldots \Phi^{k_{q-2}} \sqrt{K} \sqrt{K} \Phi^{k_{q-1}} \sqrt{I-K}\|_{\mathrm{HS}} \cdot \|\sqrt{I-K} \Phi^{k_q} \sqrt{K}\|_{\mathrm{HS}} \\
&\leq \|\sqrt{K} \Phi^{k_1} K \ldots \Phi^{k_{q-2}} \sqrt{K}\|_{\mathrm{op}} \cdot \|\sqrt{K} \Phi^{k_{q-1}} \sqrt{I-K}\|_{\mathrm{HS}} \cdot \|\sqrt{I-K} \Phi^{k_q} \sqrt{K}\|_{\mathrm{HS}} \\
&\leq \|\sqrt{K} \Phi^{k_1} K \ldots \Phi^{k_{q-2}} \sqrt{K}\|_{\mathrm{op}} \cdot k_{q-1} k_q \|\varphi\|_\infty^{k_{q-1}+k_q-2} \|\sqrt{I-K} \Phi \sqrt{K}\|_{\mathrm{HS}}^2 \\
&\leq k_{q-1} k_q \|\varphi\|_\infty^{k-2} C_2(\varphi),
\end{aligned}
$$

here we used Lemma 1, the fact that $0 \preceq K \preceq I$, $\|\Phi\|_{\mathrm{op}} = \|\varphi\|_\infty$ and the $\| \cdot \|_{\mathrm{op}}$ norm is submultiplicative. Since $k_j \leq k$ for all $1 \leq j \leq q$, using a telescoping argument gives

$$|\mathrm{Tr}[\Phi^{k_1} K \ldots \Phi^{k_q} K] - \mathrm{Tr}[\Phi^k K]| \leq q k^2 \|\varphi\|_\infty^{k-2} C_2(\varphi).$$

Hence

$$|C_k(\varphi)| \leq \sum_{q=2}^{k} \sum_{\substack{k_1,\ldots,k_q \geq 1 \\ k_1+\ldots+k_q=k}} \frac{k!}{k_1! \ldots k_q!} k^2 \|\varphi\|_\infty^{k-2} C_2(\varphi).$$

Now observe that for $k \geq 3$

$$\sum_{q=2}^{k} \sum_{\substack{k_1,\ldots,k_q \geq 1 \\ k_1+\ldots+k_q=k}} \frac{1}{k_1! \ldots k_q!} < \frac{k^k}{k!} \leq \frac{e^k}{\sqrt{2\pi k}}.$$

Thus,

$$\frac{|C_k(\varphi)|}{k!} \leq \frac{1}{\sqrt{2\pi}} e^k k^{3/2} \|\varphi\|_\infty^{k-2} C_2(\varphi).$$

$\square$

Combining all ingredients above, one can show that.

**Lemma 3.** *For $|t| \leq 1/(3\|\varphi\|_\infty)$, we have*

$$|\log \mathbb{E}[e^{t\Lambda(\varphi)}] - t\mathbb{E}[\Lambda(\varphi)]| \leq A t^2 \, \mathrm{Var}\,[\Lambda(\varphi)],$$

*where $A > 0$ is an universal constant.*

*Proof.* For $|t| \leq \frac{1}{3\|\varphi\|_\infty}$, we have

$$
\begin{aligned}
|\log \mathbb{E}[e^{t\Lambda(\varphi)}] - t\mathbb{E}[\Lambda(\varphi)]| &= \Big| \sum_{k \geq 2} \frac{C_k(\varphi)}{k!} t^k \Big| \\
&\leq \sum_{k \geq 2} \frac{|C_k(\varphi)|}{k!} |t|^k \\
&\leq |t|^2 C_2(\varphi) \Big( \frac{1}{2} + \sum_{k \geq 3} \frac{1}{\sqrt{2\pi}} e^k k^{3/2} \|\varphi\|_\infty^{k-2} |t|^{k-2} \Big) \\
&\leq |t|^2 C_2(\varphi) \Big( \frac{1}{2} + \frac{1}{\sqrt{2\pi}} e^2 \sum_{k \geq 3} k^{3/2} (e/3)^{k-2} \Big) \\
&= At^2 \operatorname{Var}[\Lambda(\varphi)],
\end{aligned}
$$

where $A > 0$ is some universal constant. $\qquad\square$

We can finish the proof of Theorem 1 as follows.

*Proof of Theorem 1.* Let $\varepsilon > 0$. We have

$$
\begin{aligned}
\log \mathbb{P}(\Lambda(\varphi) - \mathbb{E}[\Lambda(\varphi)] \geq \varepsilon) &\leq \inf_t \Big( \log \mathbb{E}[e^{t\Lambda(\varphi)}] - t\mathbb{E}[\Lambda(\varphi)] - t\varepsilon \Big) \\
&\leq \inf_t \Big( -t\varepsilon + t^2 A \operatorname{Var}[\Lambda(\varphi)] \Big)
\end{aligned}
$$

where the infimum is taken on $t \in (0, 1/(3\|\varphi\|_\infty)]$.

For $0 \leq \varepsilon \leq \frac{2A \operatorname{Var}[\Lambda(\varphi)]}{3\|\varphi\|_\infty}$, choosing

$$
t_0 = \frac{\varepsilon}{2A \operatorname{Var}[\Lambda(\varphi)]} \leq \frac{1}{3\|\varphi\|_\infty}
$$

gives

$$
\log \mathbb{P}(\Lambda(\varphi) - \mathbb{E}[\Lambda(\varphi)] \geq \varepsilon) \leq -\frac{\varepsilon^2}{4A \operatorname{Var}[\Lambda(\varphi)]}
$$

as desired. $\qquad\square$

## A.2 Proof of Theorem 2

Denote by $\Phi$ the diagonal matrix $\operatorname{Diag}(\varphi) \in \mathbb{R}^{n \times n}$. For each $t \in \mathbb{R}$, we define

$$
\mathbf{G}_t := \mathbf{I} - \exp(t\Phi) = -\sum_{k=1}^\infty \frac{t^k}{k!} \Phi^k.
$$

By the Campbell formula, we have $\mathbb{E}[e^{t\Lambda(\varphi)}] = \det[\mathbf{I} - \mathbf{G}_t \mathbf{K}], t \in \mathbb{R}$. By choosing $t \geq 0$ small enough such that $\|\mathbf{G}_t \mathbf{K}\|_{\mathrm{op}} < 1$, one can expand

$$
\log \det[\mathbf{I} - \mathbf{G}_t \mathbf{K}] = -\sum_{k=1}^\infty \frac{1}{k} \operatorname{Tr}[(\mathbf{G}_t \mathbf{K})^k].
$$

Observe that $\|\mathbf{G}_t\|_{\mathrm{op}} \leq e^{|t|\|\varphi\|_\infty} - 1 \leq 2|t|\|\varphi\|_\infty$ for all $|t| \leq 1/(3\|\varphi\|_\infty)$. From now on, we will consider $t \geq 0$ such that

$$
0 \leq t\|\varphi\|_\infty M \leq \frac{1}{3},
$$

where $M := \max(\|\mathbf{K}\|_{\mathrm{op}}, 1)$. This choice for $t$ will particularly imply that $\|\mathbf{G}_t \mathbf{K}\|_{\mathrm{op}} \leq 2/3$.

For $k = 1$, we have

$$
\begin{aligned}
-\mathrm{Tr}[\mathbf{G}_t \mathbf{K}] &= \sum_{p=1}^{\infty} \frac{t^p}{p!} \mathrm{Tr}[\Phi^p \mathbf{K}] \\
&\leq t\mathbb{E}[\Lambda(\varphi)] + \frac{t^2}{2} \mathrm{Tr}[\Phi^2 \mathbf{K}] + \sum_{p \geq 3} \frac{t^p}{p!} |\mathrm{Tr}[\Phi^p \mathbf{K}]| \\
&\leq t\mathbb{E}[\Lambda(\varphi)] + \frac{t^2}{2} \mathrm{Tr}[\Phi^2 \mathbf{K}] + \sum_{p \geq 3} \frac{t^p}{p!} \|\varphi\|_{\infty}^p \|\mathbf{K}\|_* \\
&\leq t\mathbb{E}[\Lambda(\varphi)] + \frac{t^2}{2} \mathrm{Tr}[\Phi^2 \mathbf{K}] + t^3 \|\varphi\|_{\infty}^3 \|\mathbf{K}\|_*.
\end{aligned}
$$

For $k = 2$, we have

$$
\begin{aligned}
-\frac{1}{2}\mathrm{Tr}[(\mathbf{G}_t \mathbf{K})^2] &= -\frac{1}{2} \sum_{p,q \geq 1} \frac{t^{p+q}}{p!q!} \mathrm{Tr}[\Phi^p \mathbf{K} \Phi^q \mathbf{K}] \\
&= -\frac{t^2}{2} \mathrm{Tr}[\Phi \mathbf{K} \Phi \mathbf{K}] - \frac{1}{2} \sum_{p+q \geq 3} \frac{t^{p+q}}{p!q!} \mathrm{Tr}[\Phi^p \mathbf{K} \Phi^q \mathbf{K}] \\
&\leq -\frac{t^2}{2} \mathrm{Tr}[\Phi \mathbf{K} \Phi \mathbf{K}] + \frac{1}{2} \sum_{l \geq 3} \frac{t^l}{l!} 2^l \|\varphi\|_{\infty}^l \|\mathbf{K}\|_{\mathrm{op}} \|\mathbf{K}\|_* \\
&\leq -\frac{t^2}{2} \mathrm{Tr}[\Phi \mathbf{K} \Phi \mathbf{K}] + t^3 \|\varphi\|_{\infty}^3 \|\mathbf{K}\|_* \|\mathbf{K}\|_{\mathrm{op}}.
\end{aligned}
$$

For $k \geq 3$, we observe that

$$
|\mathrm{Tr}[(\mathbf{G}_t \mathbf{K})^k]| \leq \|(\mathbf{G}_t \mathbf{K})^k\|_* \leq \|\mathbf{G}_t \mathbf{K}\|_{\mathrm{op}}^{k-3} \|\mathbf{G}_t\|_{\mathrm{op}}^3 \|\mathbf{K}\|_{\mathrm{op}}^2 \|\mathbf{K}\|_* \leq \left(\frac{2}{3}\right)^{k-3} (2t\|\varphi\|_{\infty})^3 \|\mathbf{K}\|_{\mathrm{op}}^2 \|\mathbf{K}\|_*.
$$

Thus

$$
\sum_{k \geq 3} \frac{1}{k} |\mathrm{Tr}[(\mathbf{G}_t \mathbf{K})^k]| \leq 8t^3 \|\varphi\|_{\infty}^3 \|\mathbf{K}\|_{\mathrm{op}}^2 \|\mathbf{K}\|_*.
$$

Combining all ingredients, we deduce that

$$
\begin{aligned}
\log \mathbb{E}[e^{t\Lambda(\varphi)}] &\leq t\mathbb{E}[\Lambda(\varphi)] + \frac{t^2}{2} \mathbb{V}\mathrm{ar}[\Lambda(\varphi)] + t^3 \|\varphi\|_{\infty}^3 \|\mathbf{K}\|_* (1 + \|\mathbf{K}\|_{\mathrm{op}} + 8\|\mathbf{K}\|_{\mathrm{op}}^2) \\
&\leq t\mathbb{E}[\Lambda(\varphi)] + \frac{t^2}{2} \mathbb{V}\mathrm{ar}[\Lambda(\varphi)] + 10t^3 \|\varphi\|_{\infty}^3 \|\mathbf{K}\|_* M^2.
\end{aligned}
$$

Let $\varepsilon > 0$. We have

$$
\begin{aligned}
\log \mathbb{P}(\Lambda(\varphi) - \mathbb{E}[\Lambda(\varphi)] \geq \varepsilon) &\leq \inf_t \left( \log \mathbb{E}[e^{t\Lambda(\varphi)}] - t\mathbb{E}[\Lambda(\varphi)] - t\varepsilon \right) \\
&\leq \inf_t \left( -t\varepsilon + \frac{t^2}{2} \mathbb{V}\mathrm{ar}[\Lambda(\varphi)] + t^3 \cdot 10\|\varphi\|_{\infty}^3 \|\mathbf{K}\|_* M^2 \right)
\end{aligned}
$$

where the infimum is taken on $t \in (0, 1/(3\|\varphi\|_{\infty} M)]$.

For $0 \leq \varepsilon \leq \frac{\mathbb{V}\mathrm{ar}[\Lambda(\varphi)]^2}{40\|\varphi\|_{\infty}^3 \|\mathbf{K}\|_* M^2}$, we choose

$$
t_0 = \frac{\varepsilon}{\mathbb{V}\mathrm{ar}[\Lambda(\varphi)]} \leq \frac{\mathbb{V}\mathrm{ar}[\Lambda(\varphi)]}{40\|\varphi\|_{\infty}^3 \|\mathbf{K}\|_* M^2} \leq \frac{2M\|\varphi\|_{\infty}^2 \|\mathbf{K}\|_*}{40\|\varphi\|_{\infty}^3 \|\mathbf{K}\|_* M^2} < \frac{1}{3\|\varphi\|_{\infty} M}.
$$

This choice yields

$$
\log \mathbb{P}(\Lambda(\varphi) - \mathbb{E}[\Lambda(\varphi)] \geq \varepsilon) \leq -\frac{\varepsilon^2}{2\mathbb{V}\mathrm{ar}[\Lambda(\varphi)]} + t_0^3 \cdot 10\|\varphi\|_{\infty}^3 \|\mathbf{K}\|_* M^2.
$$

Note that

$$t_0^3 \cdot 10\|\varphi\|_\infty^3 \|\mathbf{K}\|_* M^2 = \frac{\varepsilon^3}{\mathbb{V}\mathrm{ar}\left[\Lambda(\varphi)\right]^3} \cdot 10\|\varphi\|_\infty^3 \|\mathbf{K}\|_* M^2 \leq \frac{\varepsilon^2}{4\,\mathbb{V}\mathrm{ar}\left[\Lambda(\varphi)\right]}.$$

This implies

$$\log \mathbb{P}(\Lambda(\varphi) - \mathbb{E}[\Lambda(\varphi)] \geq \varepsilon) \leq -\frac{\varepsilon^2}{4\,\mathbb{V}\mathrm{ar}\left[\Lambda(\varphi)\right]}$$

as desired.

### A.3 Proof of Theorem 3

*Proof of Theorem 3.* By a scaling argument, it suffices to prove for the case $\omega_1 = \ldots = \omega_p = 1$. We have

$$\begin{aligned}
\mathbb{P}(\|\Lambda(\Phi) - \mathbb{E}[\Lambda(\Phi)]\|_2 \geq \varepsilon) &= \mathbb{P}\Big( \sum_{i=1}^p |\Lambda(\varphi_i) - \mathbb{E}[\Lambda(\varphi_i)]|^2 \geq \varepsilon^2 \Big) \\
&= \mathbb{P}\Big( \sum_{i=1}^p |\Lambda(\varphi_i) - \mathbb{E}[\Lambda(\varphi_i)]|^2 \geq \varepsilon^2 \sum_{i=1}^p \frac{\mathbb{V}\mathrm{ar}\left[\Lambda(\varphi_i)\right]}{\|\mathbb{V}(\Phi)\|_2^2} \Big) \\
&\leq \sum_{i=1}^p \mathbb{P}\Big( |\Lambda(\varphi_i) - \mathbb{E}[\Lambda(\varphi_i)]| \geq \varepsilon \frac{\sqrt{\mathbb{V}\mathrm{ar}\left[\Lambda(\varphi_i)\right]}}{\|\mathbb{V}(\Phi)\|_2} \Big).
\end{aligned}$$

For each $1 \leq i \leq p$, applying Theorem 1 gives

$$\mathbb{P}\Big( |\Lambda(\varphi_i) - \mathbb{E}[\Lambda(\varphi_i)]| \geq \varepsilon \frac{\sqrt{\mathbb{V}\mathrm{ar}\left[\Lambda(\varphi_i)\right]}}{\|\mathbb{V}(\Phi)\|_2} \Big) \leq 2\exp\Big( -\frac{\varepsilon^2}{4A\|\mathbb{V}(\Phi)\|_2^2} \Big), \forall 0 \leq \varepsilon \leq \frac{2A\|\mathbb{V}(\Phi)\|_2 \sqrt{\mathbb{V}\mathrm{ar}\left[\Lambda(\varphi_i)\right]}}{3\|\varphi_i\|_\infty}.$$

The theorem follows. □

### A.4 Proof of Theorem 4

Using (A.4), we deduce that

$$\Big| \frac{L_\mathcal{S}(f)}{L(f)} - 1 \Big| \leq \frac{1}{cn} |L_\mathcal{S}(f) - L(f)|, \quad \forall f \in \mathcal{F}.$$

This implies

$$\mathbb{P}\Big( \exists f \in \mathcal{F} : \Big| \frac{L_\mathcal{S}(f)}{L(f)} - 1 \Big| \geq \varepsilon \Big) \leq \mathbb{P}\Big( \exists f \in \mathcal{F} : \frac{1}{n} |L_\mathcal{S}(f) - L(f)| \geq c\varepsilon \Big).$$

Thus, it suffices to bound the RHS. For each $f \in \mathcal{F}$, let $V \geq \mathbb{V}\mathrm{ar}\left[n^{-1}L_\mathcal{S}(f)\right]$, we apply Theorem 1 for the linear statistic $L_\mathcal{S}(f) = \Lambda(f/\mathbf{K})$ to obtain

$$\mathbb{P}\Big( \frac{1}{n} |L_\mathcal{S}(f) - L(f)| \geq c\varepsilon \Big) \leq 2\exp\Big( -\frac{c^2\varepsilon^2}{4AV} \Big), \quad \forall 0 \leq \varepsilon \leq \frac{2AnV}{3c\|f/\mathbf{K}\|_\infty}.$$

Using $\mathbf{K}(x,x) \geq \rho m/n$, we deduce that the above inequality holds for any $0 \leq \varepsilon \leq \frac{2A\rho mV}{3c\|f\|_\infty}$.

*Proof of Theorem 4: Assuming* (A.1). We let $\mathcal{F}_{\mathrm{sym}} := \{\lambda f : |\lambda| \leq 1, f \in \mathcal{F}\}$, and let $\mathcal{B}$ be the convex hull of $\mathcal{F}_{\mathrm{sym}}$. Since $\mathcal{B}$ is a symmetric convex body in $\overline{\mathcal{F}}$, there exists a norm $\|\cdot\|_\mathcal{F}$ in $\overline{\mathcal{F}}$ such that $\mathcal{B}$ is the unit ball in $(\overline{\mathcal{F}}, \|\cdot\|_\mathcal{F})$.

Define

$$\mathcal{L}(f) := \frac{1}{n}\Big( L_\mathcal{S}(f) - L(f) \Big), \quad f \in \overline{\mathcal{F}},$$

then it is clear that $\mathcal{L}(f)$ is linear in $f$. Moreover, for any $f, g \in \overline{\mathcal{F}}$, one has

$$|\mathcal{L}(f) - \mathcal{L}(g)| = |\mathcal{L}(f - g)| = \|f - g\|_\mathcal{F} \cdot \Big| \mathcal{L}\Big( \frac{f-g}{\|f-g\|_\mathcal{F}} \Big) \Big| \leq \|f - g\|_\mathcal{F} \sup_{h \in \mathcal{B}} |\mathcal{L}(h)|.$$

For each $\delta > 0$, let $\mathcal{B}_\delta$ be a $\delta$-net for $(\mathcal{B}, \|\cdot\|_{\mathcal{F}})$. By definition of a $\delta$-net, for any $f \in \mathcal{B}$, there exists an $f_0 \in \mathcal{B}_\delta$ such that $\|f - f_0\|_{\mathcal{F}} \leq \delta$. Thus, for every $f \in \mathcal{B}$

$$|\mathcal{L}(f)| \leq |\mathcal{L}(f_0)| + |\mathcal{L}(f) - \mathcal{L}(f_0)| \leq |\mathcal{L}(f_0)| + \delta \sup_{h \in \mathcal{B}} |\mathcal{L}(h)| \leq \sup_{g \in \mathcal{B}_\delta} |\mathcal{L}(g)| + \delta \sup_{h \in \mathcal{B}} |\mathcal{L}(h)|.$$

This implies $\sup_{f \in \mathcal{B}} |\mathcal{L}(f)| \leq \frac{1}{1-\delta} \sup_{f \in \mathcal{B}_\delta} |\mathcal{L}(f)|$, $\forall 0 < \delta < 1$. In particular, choosing $\delta = 1/2$ gives

$$\sup_{f \in \mathcal{B}} |\mathcal{L}(f)| \leq 2 \sup_{f \in \mathcal{B}_{1/2}} |\mathcal{L}(f)|.$$

Therefore

$$\mathbb{P}\Big(\sup_{f \in \mathcal{B}} |\mathcal{L}(f)| \geq c\varepsilon\Big) \leq \mathbb{P}\Big(2 \sup_{f \in \mathcal{B}_{1/2}} |\mathcal{L}(f)| \geq c\varepsilon\Big) = \mathbb{P}\Big(\exists f \in \mathcal{B}_{1/2} : \frac{1}{n}|L_{\mathcal{S}}(f) - L(f)| \geq c\varepsilon/2\Big).$$

Let $N(\mathcal{B}, \|\cdot\|_{\mathcal{F}}, 1/2)$ be the $1/2$-covering number of $\mathcal{B}$, then

$$\mathbb{P}\Big(\exists f \in \mathcal{B} : \frac{1}{n}|\hat{L}_{\mathcal{S}}(f) - L(f)| \geq c\varepsilon\Big) \leq N(\mathcal{B}, \|\cdot\|_{\mathcal{F}}, 1/2) \cdot 2e^{-c^2\varepsilon^2/16AV},$$

for any

$$V \geq \sup_{f \in \mathcal{B}} \mathbb{V}\text{ar}\left[\frac{1}{n}L_{\mathcal{S}}(f)\right] \quad , \quad 0 \leq \varepsilon \leq \frac{4A\rho m V}{3c \sup_{f \in \mathcal{B}} \|f\|_\infty}.$$

Note that for a finite dimensional normed vector space, for $0 < \delta < 1$, one has

$$N(\mathcal{B}, \|\cdot\|_{\mathcal{F}}, \delta) \leq \left(\frac{3}{\delta}\right)^{\dim \overline{\mathcal{F}}}.$$

This implies

$$\mathbb{P}\Big(\exists f \in \mathcal{B} : \frac{1}{n}|L_{\mathcal{S}}(f) - L(f)| \geq c\varepsilon\Big) \leq 2\exp\left(6D - \frac{c^2\varepsilon^2}{16AV}\right). \tag{8}$$

Since $\mathcal{F} \subset \mathcal{B}$, it is clear that

$$\mathbb{P}\Big(\exists f \in \mathcal{F} : |L_{\mathcal{S}}(f) - L(f)| \geq nc\varepsilon\Big) \leq \mathbb{P}\Big(\exists f \in \mathcal{B} : |L_{\mathcal{S}}(f) - L(f)| \geq nc\varepsilon\Big). \tag{9}$$

On the other hand, for each $f \in \mathcal{B}$, there exist $0 \leq t \leq 1, |\lambda_i| \leq 1, f_i \in \mathcal{F}, i = 1, 2$ such that $f = t\lambda_1 f_1 + (1-t)\lambda_2 f_2$. Therefore

$$
\begin{aligned}
\mathbb{V}\text{ar}\left[L_{\mathcal{S}}(f)\right]^{1/2} &= \mathbb{V}\text{ar}\left[L_{\mathcal{S}}(t\lambda_1 f_1) + L_{\mathcal{S}}((1-t)\lambda_2 f_2)\right]^{1/2} \\
&\leq \mathbb{V}\text{ar}\left[L_{\mathcal{S}}(t\lambda_1 f_1)\right]^{1/2} + \mathbb{V}\text{ar}\left[L_{\mathcal{S}}((1-t)\lambda_2 f_2)\right]^{1/2} \\
&\leq t\,\mathbb{V}\text{ar}\left[L_{\mathcal{S}}(f_1)\right]^{1/2} + (1-t)\,\mathbb{V}\text{ar}\left[L_{\mathcal{S}}(f_2)\right]^{1/2} \\
&\leq \sup_{g \in \mathcal{F}} \mathbb{V}\text{ar}\left[L_{\mathcal{S}}(g)\right]^{1/2}.
\end{aligned}
$$

Moreover,

$$\|f\|_\infty = \sup_{x \in \mathcal{X}} |f(x)| \leq t \sup_{x \in \mathcal{X}} |f_1(x)| + (1-t) \sup_{x \in \mathcal{X}} |f_2(x)| \leq \sup_{g \in \mathcal{F}} \|g\|_\infty.$$

Thus,

$$\sup_{f \in \mathcal{B}} \mathbb{V}\text{ar}\left[L_{\mathcal{S}}(f)\right] = \sup_{f \in \mathcal{F}} \mathbb{V}\text{ar}\left[L_{\mathcal{S}}(f)\right] \quad , \quad \sup_{f \in \mathcal{B}} \|f\|_\infty = \sup_{f \in \mathcal{F}} \|f\|_\infty. \tag{10}$$

From (8), (9), (10), the theorem follows. $\qquad\square$

*Proof of Theorem 4:* Assuming (A.2), (A.3). We define $\mathcal{L}(\theta) := \frac{1}{n}(L_{\mathcal{S}}(f_\theta) - L(f_\theta)), \theta \in \Theta$. Then

$$\mathbb{P}\Big(\exists f \in \mathcal{F} : \frac{1}{n}|L_{\mathcal{S}}(f) - L(f)| \geq c\varepsilon\Big) = \mathbb{P}(\exists\, \theta \in \Theta : |\mathcal{L}(\theta)| \geq c\varepsilon) = \mathbb{P}(\sup_{\theta \in \Theta} |\mathcal{L}(\theta)| \geq c\varepsilon).$$

Using (A.3), we have $|L(f_\theta) - L(f_{\theta'})| \leq n\ell\|\theta - \theta'\|$ and

$$|L_{\mathcal{S}}(f_\theta) - L_{\mathcal{S}}(f_{\theta'})| \leq \ell\|\theta - \theta'\|\Big(\sum_{x \in \mathcal{S}} \frac{1}{\mathbf{K}(x,x)}\Big) \leq \ell\|\theta - \theta'\|n\rho^{-1}m^{-1}|\mathcal{S}|. \tag{11}$$

This implies $|\mathcal{L}(\theta) - \mathcal{L}(\theta')| \leq C\|\theta - \theta'\|$ a.s., for some constant $C$ depending on $B, \rho, \ell$. Let $\Gamma$ be a $\frac{c\varepsilon}{2C}$-net for $\Theta$, then

$$\sup_{\theta \in \Theta} |\mathcal{L}(\theta)| \leq \sup_{\theta' \in \Gamma} |\mathcal{L}(\theta')| + \frac{c\varepsilon}{2}.$$

Thus

$$\mathbb{P}(\sup_{\theta \in \Theta} |\mathcal{L}(\theta)| \geq c\varepsilon) \leq \mathbb{P}(\sup_{\theta' \in \Gamma} |\mathcal{L}(\theta')| \geq c\varepsilon/2)$$

We note that $|\Gamma| = O(\varepsilon^{-D})$. This completes the proof. $\qquad\square$

**Remark 6.3.** *Without the assumption $|\mathcal{S}| \leq B \cdot m$ a.s., one can continue from* (11) *as follows. Denote by $\lambda_1 \geq \ldots \geq \lambda_n \geq 0$ the eigenvalues of $\mathbf{K}$, it is known that $|\mathcal{S}| =^d X_1 + \ldots + X_n$, where $X_i \sim Ber(\lambda_i)$ are independent. Let $B > 0$, then using a multiplicative Chernoff bound for the sum of independent Bernoulli variables gives*

$$\mathbb{P}(|\mathcal{S}| > (B+1)m) = \mathbb{P}\Big(\sum_{i=1}^{n} X_i > (B+1)m\Big) \leq \exp\Big(-\frac{B^2}{B+2}m\Big).$$

*By choosing $B$ large, this event will have small probability. Meanwhile, on the event $\{|\mathcal{S}| \leq (B+1)m\}$, we can use exactly the same argument as in the proof above.*

### A.5  Proof of Theorem 5

*Proof of Theorem 5.* It suffices to show for the case $\omega_1 = \ldots = \omega_p = 1$. For each $\mathbf{f} \in \mathcal{F}$, by applying Theorem 1 and an union bound argument, we have

$$\mathbb{P}\Big(\frac{1}{n}\|L_{\mathcal{S}}(\mathbf{f}) - L(\mathbf{f})\| \geq \varepsilon\Big) \leq 2p \exp\Big(-\frac{c^2\varepsilon^2}{4AV}\Big), \quad \forall 0 \leq \varepsilon \leq \frac{2A\rho mV}{3c \max_i \|f_i\|_\infty}.$$

Using the same argument as in the proof of Theorem 4 under assumption (A.1) gives the result. $\quad\square$

### A.6  Proof of Theorem 6

*Proof of Theorem 6.* We remark that $\mathbb{V}\mathrm{ar}\big[n^{-1}L_{\mathcal{S}}(f)\big] = \mathcal{O}(m^{-(1+1/d)})$ uniformly for all $f \in \mathcal{F}$, w.h.p. in the data set $\mathcal{X}$. Hence, Theorem 6 is a direct application of Theorem 4 with $V = Cm^{-(1+1/d)}$ for some constant $C > 0$. As we discussed in the Remark 4.1, the range for $\varepsilon$ is $\mathcal{O}(m^{-1/d})$. Thus, it suffices to check the conditions on $\hat{\mathbf{K}}$. Since $\hat{\mathbf{K}}$ is a projection of rank $m$, $|\mathcal{S}| = m$ a.s. Moreover, we have $n\hat{\mathbf{K}}(x,x) = \mathbf{K}(x,x)$, which is typically of order $m$, where we used an uniform CLT result and an asymptotic for multivariate OPE kernels (see Bardenet, Ghosh, et al., 2021 for more details).

$\square$

