# OpenReview forum: "Small coresets via negative dependence: DPPs, linear statistics, and concentration"
_NeurIPS.cc/2024/Conference — NeurIPS 2024 spotlight_

### Official Review · Reviewer_818N · 2024-07-08

**Soundness:** 3
**Presentation:** 2
**Contribution:** 2
**Rating:** 6
**Confidence:** 5

**Summary:**

This paper studies corsets and aims to construct them using determinantal point processes (DPPs). The authors show that DPPs can provably outperform independently drawn corsets due to linear statistics and better concentration of DPPs.

**Strengths:**

- The topic of coreset problems is essential in machine learning.
- The paper is technically solid, providing interesting theoretical results.
- Empirical results confirm the power of DPPs in independent data.

**Weaknesses:**

- The writing can be improved. There are many theorems and remarks in the paper, which makes it confused about the main contribution. The theorems are technical but without explanations. What are the conclusions and applications of these theorems?

- Lack of coreset literature. For instance, Cohen-Addad, V., Larsen, K.G., Saulpic, D., Schwiegelshohn, C., & Sheikh-Omar, O.A. (2022). Improved Coresets for Euclidean $k$-Means; 2) Huang, Lingxiao, Jian Li and Xuan Wu. “On Optimal Coreset Construction for Euclidean $(k,z)$-Clustering.

**Questions:**

By reading the paper, I am still confused about when DPPs outperform importance sampling. Could you give some concrete examples, e.g., under what optimization problem and what dataset, what is the coreset size by DPP v.s. by importance sampling?

--
Thanks for the response. I increase my score to 5.

**Limitations:**

Yes

---

> ### Author Rebuttal · Authors · 2024-08-06
>
> Thanks for the review. Here are answers to your comments/questions that we hope will clarify things.
>
> > The writing can be improved. There are many theorems and remarks in the paper, which makes it confused about the main contribution. The theorems are technical but without explanations. What are the conclusions and applications of these theorems?
>
> We are sorry about the confusion. Since we otherwise got good marks for presentation, we are confident that we can make some limited adjustments to further clarify the role of each theorem and remark. In particular, following a suggestion of Reviewer 329X, we will explicitly include in the introduction our claim on the sample complexity of coresets with DPPs.
>
> In a nutshell, coresets built using independent samples require a cardinality in $\mathcal{O}(\epsilon^{-2})$ for a uniform multiplicative error of $\epsilon$. DPPs always come with at worst the same dependence, while *specific* DPPs come with a strictly better dependence of the cardinality to $\epsilon$. The meaning of *specific* is that the chosen DPP should come with a fast-decaying variance of linear statistics, as is the case for the DPP we introduce in Example 5. See our Remarks 4.1 and 6.1, and our answer to Reviewer 329X for more details on the impact of variance reduction on the size of the coreset.
>
> > Lack of coreset literature. For instance, Cohen-Addad, V., Larsen, K.G., Saulpic, D., Schwiegelshohn, C., & Sheikh-Omar, O.A. (2022). Improved Coresets for Euclidean-Means; 2) Huang, Lingxiao, Jian Li and Xuan Wu. “On Optimal Coreset Construction for Euclidean-Clustering.
>
> We thank the reviewer for drawing our attention to these interesting references; we will add them to our paper. We note that we see the paper as a DPP paper with a view to the coresets problem, so  this submission is a way for us to connect with the coresets-for-ML community and raise awareness on the potential of DPPs and negative dependence in that context. This explains why we limited ourselves to general references on coresets, and largely refrained from delving into the rich literature on coresets that address specific problems, such as Euclidean clustering. Our point is that a very generic DPP-based coreset construction can improve on the dependence of $m$ in $\epsilon^{-2}$ in a precise sense. We would be happy if the paper generated discussions on particular coreset settings like $k$-means clustering, for which much theoretical work is available. Comparison to dedicated literature for such applications are a natural direction for follow-up research.
>
> > By reading the paper, I am still confused about when DPPs outperform importance sampling. Could you give some concrete examples, e.g., under what optimization problem and what dataset, what is the coreset size by DPP v.s. by importance sampling?
>
> DPPs outperform importance sampling in the sense that some DPPs, like the discretized multivariate OPE of Example 5, provide randomized $\epsilon$-coresets with a cardinality $m$ that grows strictly slower as $\epsilon$ decreases; see Theorems 4 and 6, and in particular Remarks 4.1 and 6.1. Another way to say the same thing is that, under some DPPs, *the approximation error* $\epsilon$ decreases faster with $m$ than the importance sampling rate $m^{-1/2}$. This is confirmed by our experiments.
>
> The meaning of *some DPPs* above is that we need DPPs that come with a fast-decaying variance of linear statistics. This has been proved for certain DPPs, such as the discretized multivariate OPE introduced in Example 5. Note that the latter DPP does not use any information on the loss function nor the optimization algorithm used, so that our guarantee is fairly general, at the risk of the constants not being tight. The loss function only enters our guarantees through assumptions (A.1) to (A.4).
>
> Conceptually, the reason for better performance of DPPs is that the repulsion between sample points baked into the framework discourages the selection of multiple points with similar properties in the sample. DPP-based diverse samples thus cover more ground in a single sample, and can be shown to be more stable statistically. On the other hand, in importance sampling, two sample points do not “see” each other, and this can lead to oversampling in regions with higher importance scores. In the setting of Example 5, it can be shown that the variance reduction accorded by moving from uniform random sampling to importance sampling can lead to an improvement in the leading constant, whereas using an appropriate DPP can lead to an improvement in the exponent.

---

> > ### Comment · Reviewer_818N · 2024-08-14
> >
> > Thank you for your responses. The authors have addressed my questions. I would like to raise my score.

---

### Official Review · Reviewer_5TaP · 2024-07-12

**Soundness:** 4
**Presentation:** 3
**Contribution:** 3
**Rating:** 8
**Confidence:** 3

**Summary:**

The paper proves concentration inequalities for linear statistics of samples form a DPP. In particular, a guarantee for the coreset sampling problem is shown: If the coreset is sampled from a DPP then the loss over the coreset approaches the loss over the full dataset faster then when the coreset is sampled uniformly.
The superiority of DPP samples over uniform samples is demonstrated in a k-means application on toy data and MNIST data.

**Strengths:**

* proper theoretical justification of results that have been empirically observed in previous work
* The paper is well written, e.g., it contains a compact introduction to coresets that one can easily follow even if unfamiliar with coresets.
* The assumptions of the theoretical results are stated clearly and are discussed.
* informative and honest discussion section about limitations and interesting future work directions
* The results generalize beyond the coreset problem and could potentially be interesting for other DPP application areas, too.
* The results also apply to non-symmetric kernels.

**Weaknesses:**

* The markers in Figure 1 (a) and 2 (a) are different sizes, but I can't find information on what a marker's size encodes. I would appreciate a clarification.
* The comparison to the related work "Tremblay et al 2019" (see line 52-55) could be a little bit more detailed. It's just mentioned that this other paper also contains theoretical results for the same/ a similar setting, but not what their nature is and how they differ from the results in the present paper (I don't doubt they do, but I think 1-2 sentences more on this would be useful).
* The "stratified" baseline seems much stronger than the uniform baseline and the DPP sampler—at least, where it is straightforward to apply. Other heuristical sampling methods that encode repulsiveness between data items without being DPPs might also be preferable in large-scale machine learning applications, including coreset problems. The practical relevance might be limited, but I don't consider this a major issue as the paper is of a theoretical nature.

**Questions:**

Remark 3.3 mentioned that the assumption on a set's cardinality holds for most kernels of interest. Are there more examples other than the projection kernels? I assume it holds for the Gaussian kernel as it is used in the experiments? Are there examples of common kernels where the assumption does not hold?

**Limitations:**

yes, there is a dedicated limitations sections that I perceive as accurate

---

> ### Author Rebuttal · Authors · 2024-08-06
>
> Thanks for the positive review. Here are answers to some of your comments/questions.
>
> > The markers in Figure 1 (a) and 2 (a) are different sizes, but I can't find information on what a marker's size encodes. I would appreciate a clarification.
>
> The size of a marker placed at $x$ is proportional to the corresponding weight $1/K(x,x)$ in the estimator of the average loss. Equivalently, the marker size is inversely proportional to the marginal probability of $x$ being included in the DPP sample. We will clarify this in the final version of the paper.
>
> > The comparison to the related work "Tremblay et al 2019" (see line 52-55) could be a little bit more detailed. [...] I think 1-2 sentences more on this would be useful).
>
> We will add a brief discussion to clarify the differences. The most important thing for our paper is that Tremblay et al 2019 only prove that DPP-based coresets of accuracy $\epsilon$ should have cardinality $\mathcal{O}(\epsilon^{-2})$, which is the same rate as independent samples. This falls short of demonstrating that repulsiveness of DPPs leads to an improvement in the rate of convergence in sampling tasks, as believed from physical heuristics. This provides a natural   Besides our generic concentration inequalities, our key contribution is that we show that appropriately chosen DPPs can lead to significantly smaller coresets, i.e. of cardinality $\mathcal{O}(\epsilon^{-\gamma})$ with $\gamma > 2$ as $\epsilon$ goes to zero.
> Tremblay et al. (2019) have a number of other interesting results in their paper, for instance that for any DPP with an Hermitian kernel that has for diagonal the sensitivity, the cardinality of the DPP coreset is smaller than i.i.d. sampling proportionally to the sensitivity. This is another hint that repulsiveness helps, but it is not a direct proof of the heuristic that smaller DPP coresets work. A key contribution of ours is that we quantify how small DPP coresets can be, and we show that they can break the $\mathcal{O}(\epsilon^{-2})$ barrier of independent samples.
>
> > The "stratified" baseline seems much stronger than the uniform baseline and the DPP sampler—at least, where it is straightforward to apply. Other heuristical sampling methods that encode repulsiveness between data items without being DPPs might also be preferable in large-scale machine learning applications, including coreset problems. The practical relevance might be limited, but I don't consider this a major issue as the paper is of a theoretical nature.
>
> Actually, it is not hard to show that stratified sampling as we implement is a DPP, although it is rarely introduced that way. But we agree that there are plenty of useful repulsive methods and heuristics that are not DPPs. However, the stochastic dependencies in other repulsive or negatively dependent heuristics are often too complicated to understand from a theoretical point of view. As such, they come with very little mathematical guarantees. We see DPPs as a tractable tool to mathematically quantify the power of negative dependence in sampling applications.
>
> > Remark 3.3 mentioned that the assumption on a set's cardinality holds for most kernels of interest. Are there more examples other than the projection kernels? I assume it holds for the Gaussian kernel as it is used in the experiments? Are there examples of common kernels where the assumption does not hold?
>
> There is no Remark 3.3, but you likely mean Remark 4.3 or 6.3, which both discuss cardinality. The assumption $\vert\mathcal{S}\vert \leq Bm$ a.s. is equivalent to the kernel having at most $Bm$ nonzero eigenvalues; this includes DPPs with projection as well as many non-projection kernels. Note that the assumption is here to simplify the proof for the reader who focuses on projection kernels, but the assumption is easy to avoid for a generic Hermitian kernel, thanks to a control on the tail $\mathbb{P}(\vert \mathcal{S}\vert\geq (B+1)m)$ and our assumption that the marginal probability of seeing any item is lower bounded, as mentioned in Remarks 4.3 and 6.3. The case of a DPP with Gaussian kernel and Gaussian reference measure, where the number of nonzero eigenvalues is infinite, is covered by that extension.
>
> For the Gaussian kernel in the experiments, the situation is slightly different. We indeed include $m$-DPPs with Gaussian kernels in our experiments, but an $m$-DPP is not a DPP. Strictly speaking, our proof thus does not cover $m$-DPPs, and we rather include $m$-DPPs in our experiments for comparison to previous work. Note however that $m$-DPPs are conditional DPPs (conditioned on the number of points), and we could easily extend the proof of our concentration inequalities by a simple conditioning argument, if needed for the sake of exhaustiveness.

---

> > ### Comment · Reviewer_5TaP · 2024-08-12
> >
> > Thank you for the clarifications. They make sense to me. I keep my positive score.

---

### Official Review · Reviewer_koaH · 2024-07-13

**Soundness:** 3
**Presentation:** 3
**Contribution:** 3
**Rating:** 6
**Confidence:** 3

**Summary:**

This paper presents a study on the use of Determinantal Point Processes (DPPs) for constructing coresets in machine learning tasks. DPPs are random configurations of points with negative dependence, making them suitable for subsampling tasks like minibatch selection or coreset construction. Therefore, it is natural to ask can DPP-based coresets have smaller size than the independent-sampling-based coresets. The paper answers the question, demonstrating that DPP-based method outperforms.

To achieve this, the author provide a new understanding of coreset loss as a linear statistic of the random point set. Then they connect the coreset construction to the concentration inequalities of the linear statistic and show that a suitable DPP yields a coreset of size $o(\varepsilon^{-2})$.

**Strengths:**

- New concentration inequalities for linear statistics of DPPs are presented, which are applicable to general non-projection and non-symmetric kernels.
- The DPPs-based coresets have a smaller theoretical size than the independent-sampling-based coresets.
- The paper introduces the study of coresets for vector-valued objective functions, a topic that holds independent interest.

**Weaknesses:**

see the questions

**Questions:**

- What is the time complexity of DPP-based coreset construction?
- The paper proposes some assumptions on the DPPs. Are these assumptions fair compared to previous methods?

---

> ### Author Rebuttal · Authors · 2024-08-06
>
> Thanks for the positive review. Here are answers to your questions.
>
> > What is the time complexity of DPP-based coreset construction?
>
> In general, sampling a DPP of cardinality $m$ among a ground set of size $n$ is $O(nm^2)$ provided the kernel matrix has been diagonalized beforehand. Depending on how the kernel is specified, the diagonalization preprocessing can add up to $O(n^3)$ operations. In practice, randomization and low-rank approximation techniques can be used to take down these costs for large-scale applications; see e.g. [Kulesza and Taskar, FTML 2012](https://www.nowpublishers.com/article/Details/MAL-044).
>
> Given that coreset guarantees are uniform bounds for entire function classes, the same coreset once sampled can give accurate approximation for a range of functions of interest for a particular application, and thus DPP sampling is essentially a one-time preprocessing overhead.
>
> On a related note, leveraging the fact that the coreset-based approximant is a linear statistic, the sampling problem may be reduced to that of sampling a real-valued random variable via its Laplace transform, cf. [Bardenet et al., Arxiv](https://arxiv.org/abs/2007.04287).
>
> It may be noted that in many modern coreset applications, the functional evaluation is often computationally very expensive, and the cost of coreset sampling can be secondary to that. A typical example would be the computation of the output of a large-scale neural network, or its training dynamics where the gradient is computed via a backpropagation through the network. Another instance is that of large-scale conditional random field models. In such situations, it is imperative to reduce the number of function evaluations, and smaller coresets are very helpful in achieving that objective.
>
> > The paper proposes some assumptions on the DPPs. Are these assumptions fair compared to previous methods?
>
> Our concentration results are very generic, and we even consider DPPs with non-symmetric kernels. In the application to coreset construction, we make very minimal assumptions on the DPP that is used. In fact, we only bound from below the marginal probability of seeing any item of the ground set in the coreset; see Remark 4.2. This seems reasonable to us: since our goal is to estimate a mean over the $n$ data points, there should be a chance, even minute, of sampling each data point.
>
> Most of our assumptions are on the complexity of the space of queries $\mathcal F$. However, as demonstrated in our paper, these assumptions cover most of the learning tasks for which coresets have been studied: $k$-means, linear regression, etc, and also some important spaces of test functions of other fields (e.g. band-limited functions in signal processing).

---

> > ### Comment · Reviewer_koaH · 2024-08-14
> >
> > Thank you for your responses. The authors have addressed some of my questions. I would like keep my positive score.

---

> > > ### Author Response · Authors · 2024-08-14
> > >
> > > Thank you very much for your positive response. Please let us know if you have further questions, or would like to have additional clarifications; we would be happy to furnish further details.

---

### Official Review · Reviewer_329X · 2024-07-16

**Soundness:** 3
**Presentation:** 3
**Contribution:** 3
**Rating:** 7
**Confidence:** 2

**Summary:**

This work improves on the concentration bounds on DPP for coresets by explicitly relying on the fact that loss based coresets are linear functionals. To this end, this paper also add the coreset results for Non-symmetric kernels, (including additive coresets) expanding on the previous results on DPP Coresets (which were solely in the hermitian regime). Paper compares against other sampling based theoretical coresets (including DPPs) for approximation errors.

**Strengths:**

I like the fact that this paper extended the DPP bounds to Non-symmetric kernels. Overall the paper is not difficult to parse through, despite many theoretical statements.

**Weaknesses:**

I cannot exactly pin down the weakness in this work since I'm not expert in this area, however, I'd like to point out a few things that can help to understand this work better.

- When going from lipschitz (Peres and Pemantle) to linear functional concentration, what exactly changes which leads to the better bounds?

- Can the authors have a side by side comparision of the sample complexity for the previous existing works (DPP for coresets paper), in a tablular form to understand the contributions of this work better?

- What is an intuitive explanation of the fact that the range of $\epsilon$ got tighter in non-symmetric case?

- Can there be chaining related improvements to further improve the provided bounds, as done in Bhatt and Bilmes 2021?

- Is it possible to provide some experimental results with different choice of kernels including non-symmetric?

- Writing Suggestion: It might improve readability of $\|\varphi\|_{\infty}$ is replaced by some constant bounding the loss function.

References:
- Tighter m-DPP Coreset Sample Complexity Bounds, Bhatt and Bilmes 2021. Subset ML at ICML'21

**Questions:**

Refer to the weakness.

**Limitations:**

Refer to the weakness.

---

> ### Author Rebuttal · Authors · 2024-08-06
>
> Thanks for the positive review! Here are answers to your comments.
>
> > When going from Lipschitz (Peres and Pemantle) to linear functional concentration, what exactly changes which leads to the better bounds?
>
> The Pemantle-Peres results aim to address general Lipschitz functions in several variables. Roughly speaking, one has to pay for the generality by having a much cruder result. A key point we make is that, in many practical situations, functionals of DPPs that are of interest have a linear structure. Therefore it is natural to develop a theory of concentration of linear functionals of a DPP, as we do in this paper. Unlike the general Lipschitz case, the Laplace transform of linear statistics of DPPs is explicitly given by a Fredholm determinant. This is the starting point of the derivation of a tighter concentration result. Another advantage is that careful work around this Fredholm determinant allows keeping in the bound the variance of the linear statistic under scrutiny, like in the classical Bernstein inequality. In that view, the result of Pemantle and Peres is more similar to the Hoeffding inequality.
>
> > Can the authors have a side by side comparison of the sample complexity for the previous existing works (DPP for coresets paper), in a tabular form to understand the contributions of this work better?
>
> We will mention this early on in the introduction when we list our contributions. Right now, the sample complexity is indeed a bit hidden in our Remark 4.1. Essentially, the best (multiplicative) $\epsilon$-coresets built with independent samples are of size $m=\mathcal{O}(\epsilon^{-2})$. Using the general result of Pemantle and Peres, Tremblay et al. (the “DPPs for coresets” paper) provided a bound with the same dependence in $\epsilon$ for DPPs. On our side, we show that we can actually take $m=\mathcal{O}(\epsilon^{-2/(1+\delta)})$, where $\delta$ depends on the variance of the subsampled loss under the considered DPP. In particular, for the discretized multivariate OPE introduced in Example 5, any $\delta \in(0,1/d)$ works, where data are assumed to live in $\mathbb{R}^d$ (in fact, in practical terms we recommend applying the algorithm there on dimensionally reduced data, and the $d$ here can be taken to be this reduced dimension).  This shows that if the DPP is suitably chosen, the dependence of the coreset size in $\epsilon$ is provably better than for independent coresets.
>
> A table summarizing our sample complexity could look like this.
>
> | Independent sampling | Generic DPPs | Discretized multivariate OPE |
> |---|---|---|
> | $\mathcal{O}(\epsilon^{-2})$ | $\mathcal{O}(\epsilon^{-2/(1+\delta)})$ for $\delta \geq 0$ based on the DPP | $\mathcal{O}(\epsilon^{-2/(1+1/d-\eta)})$ for any $\eta>0$. |
>
>
> > What is an intuitive explanation of the fact that the range of $\epsilon$ got tighter in the non-symmetric case?
>
> It is not so clear to us that the range gets tighter. The nuclear norm of the kernel is roughly $m$, so that everything depends again on the variance of the linear statistic. Much less is known on the latter for non-symmetric kernels.
>
> In fact, although the bounds look similar in the symmetric and non-symmetric cases, the non-symmetric case forced us to change our proof technique, leading to a different range for $\epsilon$. In general, for symmetric kernels, an elaborate spectral theory is available and generally finer analysis (e.g., involving algebraic cancellations) is possible. A case in point would be the Christoffel-Darboux theory that allows for the improved bounds in Example 5.
>
> > Can there be chaining related improvements to further improve the provided bounds, as done in Bhatt and Bilmes 2021?
>
> Thanks for bringing this interesting paper to our attention. There could be a small improvement in a smart chaining argument, but it is very unlikely that we can substantially improve the rate in the dependence of $m$ in $\epsilon$. In [Bhatt and Bilmes 2021], they indeed gain a log term that was implicitly there in [Tremblay et al. 2019]; this definitely makes for a cleaner result, but does not substantially change the rate.
>
> > Is it possible to provide some experimental results with different choice of kernels including non-symmetric?
>
> In our experiments, we do explore several choices of kernels. These kernels are all symmetric in nature. While our concentration results are generally applicable, our experimental investigations in the paper focus on the problem of coreset construction. The setting of non-symmetric kernels is not very natural to the problem of sampling coresets. Generally speaking, non-symmetric kernels are much less tractable, analytically and algorithmically. Therefore, in the problem of DPP-based coreset construction, where the kernel is our choice, one nearly always goes for a symmetric kernel to produce the desired repulsive effects among sample points and the attendant variance reduction. In particular, the fast decay of the variance of linear statistics is key in deriving *small* coresets from our concentration inequality. To our knowledge, there is no known non-symmetric DPP with a fast-decaying variance of linear statistics, and we thus do not have a clear non-symmetric candidate DPP for comparing coresets.
>
> A key application of non-symmetric kernels is in situations where it is desirable that the mutual repulsion between points is violated at certain locations; cf [Gartrell et al. 2019](https://arxiv.org/abs/1905.12962), or  [Poulson 2019](https://arxiv.org/abs/1905.00165) and references therein. In such problems, it is of natural interest to obtain confidence bands for the outputs of the non-symmetric DPP based methods, and our concentration results for non-symmetric DPPs are the first ones to provide such guarantees. The experimental analysis related to our concentration results for the non-symmetric case are thus better suited to a different work which focuses on such relevant applications of non-symmetric kernels.

---

> > ### Comment · Reviewer_329X · 2024-08-08
> > **Thanks for the rebuttal, increasing score.**
> >
> > Thanks for responding to my questions! I believe this discussion can help the manuscript. I am raising my scores.

---

### Author Rebuttal · Authors · 2024-08-06

We thank the reviewers for their careful reading of the paper, and their detailed comments, to which we have responded point by point below. We would like to take this opportunity to elaborate on the fact that our submission is appropriately viewed in the context of a wider programme of leveraging negative dependence as a toolbox for machine learning. In particular, this enables us to exploit the diversity of samples generated using a DPP to provide more representative and more parsimonious summaries of datasets (or approximations to functions of datasets). While this is motivated by physical heuristics coming from quantum and statistical mechanics, establishing this as a principled learning paradigm with robust and rigorous guarantees has proven to be a much more difficult task, with only a limited number of use-cases to support more extensive experimental evidence. Our work contributes to this overall programme by addressing the problem of sampling coresets.

From this perspective, for our paper, the coresets problem is an important application domain wherein we establish the effectiveness of negative association-based methods (in contrast with, e.g., a paper that is situated deep within the coresets literature). As such, our evaluations and comparison benchmarks are also principally based on literature that has attempted to accomplish this goal (e.g., Tremblay et al. 2019). On a related note, discussion of the nuances of the coresets literature in specific settings (such as clustering or regression) in our paper are somewhat muted, especially in view of the limited space available in a NeurIPS submission. Having established foundational guarantees for DPPs as a coresets generating mechanism in this paper, it would be natural to explore in future works the performance of DPP-based coresets in such specific applications. Such future works would be natural venues where detailed comparisons to established benchmarks in those particular settings can be undertaken. In contrast, the present work focuses on very general, foundational guarantees for an approach based on negative association, and is ideally perceived and evaluated in that framework.

---

### Decision · Program_Chairs · 2024-09-25

**Decision:**

Accept (spotlight)

**Comment:**

This paper introduces significant new theoretical results showing that coresets sampled using DPPs can asymptotically outperform those sampled independently, and demonstrates the effect in simple empirical settings. The results are based on novel concentration inequalities for linear statistics of DPPs, which could also be of independent interest.

Following the author rebuttal, all reviewers felt that the paper deserves acceptance. The writing is clear (particularly given its technical density), the problem is important, and the results themselves are clean and general (including DPPs with non-symmetric kernels). Please consider the suggestions made by reviewers when preparing the final version of the paper.